# An enhanced decision-making framework for predicting future trends of sharing economy

**Qiong Wu[1], Xiaoxiao Tang[1], Rongjie Li[2]\*, Lei Liu[3], Hui-Ling Chen[4]\***

**1** School of Marxism, Wenzhou University, Wenzhou, China, **2** Wenzhou Business College, Wenzhou, China, **3** College of Computer Science, Sichuan University, Chengdu, Sichuan, China, **4** College of Computer Science an Artificial Intelligence, Wenzhou University, Wenzhou, China

\* tony63277418@163.com (RL); chenhuiling.jlu@gmail.com (H-LC)

**Data Availability Statement:** The dataset analyzed during the current study is available at https://github.com/Hollow123e/Sharing-Economy-Dataset.git.

## Abstract

This work aims to provide a reliable and intelligent prediction model for future trends in sharing economy. Moreover, it presents valuable insights for decision-making and policy development by relevant governmental bodies. Furthermore, the study introduces a predictive system that incorporates an enhanced Harris Hawk Optimization (HHO) algorithm and a K-Nearest Neighbor (KNN) forecasting framework. The method utilizes an improved simulated annealing mechanism and a Gaussian bare bone structure to improve the original HHO, termed SGHHO. To achieve optimal prediction performance and identify essential features, a refined simulated annealing mechanism is employed to mitigate the susceptibility of the original HHO algorithm to local optima. The algorithm employs a mechanism that boosts its global search ability by generating fresh solution sets at a specific likelihood. This mechanism dynamically adjusts the equilibrium between the exploration and exploitation phases, incorporating the Gaussian bare bone strategy. The best classification model (SGHHO-KNN) is developed to mine the key features with the improvement of both strategies. To assess the exceptional efficacy of the SGHHO algorithm, this investigation conducted a series of comparative trials employing the function set of IEEE CEC 2014. The outcomes of these experiments unequivocally demonstrate that the SGHHO algorithm outperforms the original HHO algorithm on 96.7% of the functions, substantiating its remarkable superiority. The algorithm can achieve the optimal value of the function on 67% of the tested functions and significantly outperforms other competing algorithms. In addition, the key features selected by the SGHHO-KNN model in the prediction experiment, including " Form of sharing economy in your region " and " Attitudes to the sharing economy ", are important for predicting the future trends of the sharing economy in this study. The results of the prediction demonstrate that the proposed model achieves an accuracy rate of 99.70% and a specificity rate of 99.38%. Consequently, the SGHHO-KNN model holds great potential as a reliable tool for forecasting the forthcoming trajectory of the sharing economy.

## 1. Introduction

In the year 1978, Marcus Felson, an esteemed Sociology professor at Texas State University, and Joe L. Spaeth, a professor specializing in Sociology at the University of Illinois, introduced

**Funding:** The author(s) received no specific funding for this work.

**Competing interests:** The authors have declared that no competing interests exist.

the notion of the sharing economy in their scholarly publication, marking its inaugural mention [1,2]. It is a market-based model in which the supplier of resources provides idle resources or skills to the demander through the technology platform and charges a certain fee [3]. The effective allocation of resources is achieved after integrating the technology platforms. Specifically, it can combine various business models such as sharing, renting and swapping together, which is different from traditional business models and signals the emergence of a new economic model [4]. To some extent, the sharing economy model enables the optimal allocation and use of relevant resources. Unlike other economic models, the sharing economy model is an inevitable product of the internet era, as consumers are disadvantaged in the defense of their rights by businesses, dissatisfied with the asymmetry of transaction information and lack of trust in social transaction mechanisms, combined with the full spread of internet technology and continued innovation, provide the prerequisites for the birth of the sharing economy model [5]. Although the concept of "sharing economy" has been around for a long time, the academic community has not yet fully agreed on a definition of the term. Academics agree that platforms are important foundations for the existence and development of sharing economy [6,7]. A quality platform enables effective exchange in the sharing economy, allowing consumers to consume higher-quality products at a lower cost. Meanwhile, the good reputation of the platform is transferred to the merchant, increasing consumer trust in the merchant and effectively facilitating trading activities. Overall, the sharing economy can bring unused items back into circulation without changing ownership, enabling efficient allocation of resources [8].

From a certain point of view, the application of the sharing economy can achieve the Pareto optimum, which can promote the creation of benefits for the sharing platform and the participating entities. Secondly, the elimination of information asymmetries can be used to achieve free market matching, so that idle resources can be utilized to the greatest extent while factor costs are effectively controlled [9]. Finally, the start-up costs of internet platforms can be effectively controlled under the influence of the sharing economy, and a more low-risk approach can be adopted to promote the stable operation of asset-light. Nowadays, sharing economy has been popular in many countries around the world. According to forecasts by relevant institutions, the global market of sharing economy for major industries will grow from $14 billion in 2014 to $335 billion by 2025. In recent years, advances in technology and the growing popularity of mobile communication devices and the internet have led to the widespread emergence of the sharing economy phenomenon, which has also brought a huge impact on traditional consumption models [10,11]. On the one hand: it is more adaptable to the consumption experience and consumption needs of the epidemic situation and networked development; that is, sharing economy model can replace the previous intermediary model with producer-consumer transactions, i.e. replacing the previous employment relationship with a contractual relationship between the consumer and the platform, giving consumers a better service experience. Besides, the price of products and services in the sharing economy is significantly lower than in the traditional economy, so total social demand can be significantly increased by sharing economy. However, on the other hand, due to its imperfect development, the implementation of sharing economy involves product food testing, third-party payment supervision, information security checks, logistics supervision and other means, and its regulatory bodies include industry and commerce, telecommunications, finance, quality inspection and public security [12,13]. And looking at the current development of the sharing economy model, its activity still lacks regulatory bodies. The regulatory bodies have not yet made a clear delineation of powers and responsibilities, making the regulation of the sharing economy a phenomenon of overlapping powers and responsibilities and regulatory gaps [14,15]. In addition to this, there are also many problems such as low-price dumping, monopoly agreements, unfair

competition and big data "killing", which are still difficult to be accepted for some users. In short, as a new trend, the acceptance of the sharing economy among the population is still unknown. Therefore, it is necessary to analyze the data to understand in detail the level of awareness and acceptance of the sharing economy. This will not only help to anticipate the future development of sharing economy and adjustment of relevant policies and regulations but will also help to further guide the benign operation of the sharing economy [2,16].

Recently, artificial intelligence technology has developed rapidly and applied successfully in predicting the future development of sharing economy. Klos et al. [17] specifically address the issue of sharing economy. Pilot work was carried out on the phenomenon of collective consumption of Polish consumers by means of a questionnaire to analyze the influence of sharing economy on consumers. Perles et al. [18] aimed to investigate the extent to which sharing economy affects the tourism industry. Machine learning techniques were used to solve practical problems in the tourism industry to further analyze the potential impact of the sharing economy on the tourism industry. Chen et al. [19] used structural equation modeling on a questionnaire survey of 90 sharing economy companies to explore the sustainability of sharing economy. Ackermann et al. [20] examine consumer use of sharing economy platforms from a legitimacy perspective, exploring the impact of consumer attitudes and behavioral intentions towards the accommodation sector in the context of sharing economy. Cheng et al. [21] studied the carbon footprint of Airbnb hosts in Sydney using peer-to-peer sharing on the Airbnb platform and analyses the notion that sharing economy facilitates the utilization of underutilized resources. This study contributes to the sustainability of peer-to-peer accommodation and sharing economy more generally. Ferreri et al. [22] explored the interrelationship between sharing platform economy companies and governments under the background of sharing economy, analyzing the trade-offs between corporate and public interests. Zemla et al. [23] explored the role of sharing economy in urban economic planning and analyzed the planning options made for the development and implementation of sharing economy in specific cities.

Artificial intelligence technology has experienced significant advancements in the past few years, witnessing a remarkable pace of progress, and machine learning algorithms have been of great practical value in both academia and industry. With the exponential growth and intricacy of big data, the conventional machine learning algorithms designed for small-scale data have become inadequate for addressing the multitude of challenges presented by big data applications [24,25]. Therefore, the study of machine learning algorithms in the big data environment has become a common topic of interest for both academia and industry and has wide application to the problem of predicting the future development of the sharing economy. Kim et al. [26] used Random Forest, XGBoost and LightGBM models to predict the demand for shared bicycles and integrated the predictions into the company's business operations to better serve the needs of customers. Wang et al. [27] constructed a multi-relational network to analyze the value of cooperation in the sharing economy by using different machine learning classification models and feature sets to predict consumers' purchase behavior on the platform. Tornberg et al. [28] used a machine learning approach to classify picture profiles provided by landlords of shared rentals to assess the gap between race and gender income and to further explore the impact of sharing economy. Shokoohyar et al. [29] use multiple machine learning models (support vector machines, plain Bayes and neural networks) to predict the highest return rental strategy for a specific property in Philadelphia in the context of sharing economy based on data from 2163 properties. Nadeem et al. [30] cut through the theoretical issues of consumer participation in sharing economy platforms from a multi-dimensional perspective, using marketing and business theory literature to analyze consumers' value co-creation orientation to predict future trends in sharing economy. Jiang et al. [31] conducted a study on bike-

sharing. A deep learning model is used to predict the most likely destination for each user to further satisfy consumer demand to explore the influence of sharing economy on people.

Many swarm intelligence algorithms were proposed one after another in the early 1990s, such as hunger games search (HGS) [32], colony predation algorithm (CPA) [33], slime mould algorithm (SMA) [34,35], Harris hawks optimization (HHO) [36], Runge Kutta optimizer (RUN) [37], weighted mean of vectors (INFO) [38], and rime optimization algorithm (RIME) [39]. They have achieved exciting results on combinatorial optimization problems with NP-hard characteristics that are difficult to be handled by traditional optimization algorithms [40,41]. It has garnered significant attention from the academic community and has been rapidly applied in various practice environments, particularly achieving enormous success in engineering applications. For example, they have gained wide application and achieved good results in areas such as combinatorial optimization [42,43], data mining [44,45], energy scheduling [46,47], medicine [48,49] and image classification [50,51], bankruptcy prediction [52], economic emission dispatch [53], feature selection [54–58], numerical optimization [59–61], scheduling optimization [62,63], multi-objective optimization [64], large-scale complex optimization [65], global optimization [66–70], feed-forward neural networks [71], and target tracking [72]. Therefore, to better explore the future development trend of sharing economy, this study proposes an effective prediction model for the future development trend of sharing economy. This model combines KNN with the improved Harris Hawk optimization algorithm, named SGHHO-KNN model. In the proposed algorithm, two novel mechanisms are introduced to the original HHO to improve its disadvantages. First, embedding an enhanced simulated annealing mechanism to address the tendency of the original HHO to fall into local optima. This mechanism generates new solution sets with a certain probability, enhancing the chance of the basic HHO escaping from the local optimum and thus strengthening the capability of the algorithm in global search. Secondly, by incorporating the principles of the Gaussian bare bone mechanism, the SGHHO algorithm effectively navigates the transition from exploration to exploitation phases through a refined adjustment process. This ensures both population diversity and enables the algorithm to achieve enhanced accuracy in problem-solving. Lastly, to further enhance classification performance, the SGHHO algorithm is synergistically integrated with a KNN classifier, which facilitates the evaluation of feature subsets and ultimately leads to improved classification outcomes. Simultaneously, a comprehensive set of optimization experiments was undertaken using the well-established IEEE CEC 2014 benchmark functions. The simulation outcomes unequivocally demonstrate the remarkable superiority of the proposed algorithm over the original HHO algorithm, exhibiting a significantly higher performance level on 96.7% of the functions, while also exhibiting improved stability. Moreover, to thoroughly examine the key factors influencing the future trajectory of the sharing economy, it is imperative to undertake comparative experiments involving SGHHO-KNN and other state-of-the-art algorithms. The empirical findings demonstrate that SGHHO-KNN outperforms conventional approaches across all four evaluated metrics, showcasing superior classification accuracy and enhanced stability. Notably, the prediction outcomes reveal exceptional performance, with the proposed model achieving an impressive accuracy rate of 99.70% and a remarkable specificity rate of 99.38%. Ultimately, the primary objective of this investigation is to conduct an in-depth analysis of the influence of the sharing economy on individuals, consequently providing valuable insights into the anticipated future trajectory of this economic phenomenon.

This work has made the following contributions:

- An improved simulated annealing mechanism (ISA) and a Gaussian bare bone strategy (GB) are introduced and applied to the HHO to enhance its optimization performance.

- The superior performance of SGHHO is effectively validated through comprehensive benchmark function experiments.

- A SGHHO-KNN-based model is developed to predict the future development of sharing economy.

- Effective forecasting of future trends in the sharing economy is realized and relevant key features are filtered out.

This is the organization of this study. Section 2 gives the basic principles of the HHO algorithm; Section 3 introduces SGHHO proposed in this paper, together with the improved simulated annealing mechanism and the Gaussian bare bone variation strategy introduced. Section 4 introduces the SGHHO-KNN model; Section 5 evaluates the optimization performance of SGHHO on several benchmark functions; Section 6 uses the SGHHO-KNN model to predict the future development of the sharing economy. Finally, summing up the research results of this work and giving the directions of future work.

## 2. Related works

### 2.1. Literature review

The research shows that the internal parameters in machine learning models and the feature space and sample space in big data have an important influence on the performance of classifiers. In the last decade or so, swarm intelligence algorithms have shown good benefits in solving such problems [73–76]. Furthermore, swarm intelligence algorithms exhibit notable characteristics of randomness and efficiency, encompassing a range of deterministic and stochastic techniques that find wide application in diverse optimization problems and practical scenarios, effectively harnessing the power of collective intelligence. Examples include optimization problems [77–80], traveler problems [81], complex network problems [82,83], path planning problems [84,85], and real-time detection problems [86]. Within the scope of this investigation, a refined variation of the Harris Hawks algorithm is introduced, leveraged to forecast forthcoming patterns in the advancement of the sharing economy. Originating from the pioneering work of AliAsghar Heidari and SeyedaliMirjalili in 2019, the Harris Hawks Optimization (HHO) algorithm represents a biomimetic intelligent optimization technique. The algorithm mimics the Harris Hawk predation characteristics and combines Levy Flights to achieve solutions to complex multidimensional problems. The algorithm is similar to common meta-heuristic algorithms, inspired by the habits of nature's animals. It achieves global optimization algorithms by imitating the group hunting, raiding and siege strategy of the Harris Hawk. Since its introduction, the Harris Hawk algorithm has been used to solve optimization problems in a variety of fields and has achieved good results. Literature [87] introduced the HHO algorithm with chaotic search and opposition learning to perform parameter searches for PV cells and modules. The literature [88] used the HHO algorithm to analyze the stability of mass slope problems and to improve the accuracy of predictions. Literature [89] applied the HHO to in-vehicle location services for intelligent transportation systems to reduce the delay in the delivery of emergency data in in-vehicle networks to enhance the location rate. The literature [90] used the improved adaptive Harris Hawk algorithm for obtaining the optimal queue for the 2D grey scale gradient method and used this gradient method for image processing, possessing better results compared to other multi-stage min-valorization methods. The literature [91] used HHO for industrial safety. Using HHO for augmented artificial neural networks (ANN), a hybrid model ANN-HHO model was proposed for predicting the scour depth of spillways by dam outflow water and compared with two other hybrid models ANN-GA and ANN-PSO, showing the superiority and effectiveness of the hybrid model.

## 2.2. K-nearest neighbor classifier

The KNN algorithm is a simple and effective classification algorithm. It is based on the idea of a nearest neighbor, i.e. the class of a new sample is determined by the nearest samples of a known class. The fundamental concept underlying the K-nearest neighbors (KNN) algorithm involves the computation of the distance between the sample under classification and all the training samples, utilizing a distance metric. By identifying the k closest neighbors to the sample in question [92], a majority vote is subsequently employed to ascertain the category to which the sample belongs, based on the categories assigned to these nearest neighbors. Many research works have been done on this algorithm by scholars at home and abroad. For example, Wai Lam et al. [93] from the Chinese University of Hong Kong combined the KNN method with a linear classifier and achieved better classification results, with an accuracy of over 80% at a recall rate close to 90%. The literature [94] investigated the asymptotic nature of the K-nearest neighbor estimate of the regression function and obtained the asymptotic normality of the K-nearest neighbor estimate of the regression function and the coincidence of its Bootstrap statistic. The literature [95] establishes an efficient search tree for the nearest neighbor algorithm and improves the query rate. In the literature [96], an iterative nearest neighbor method is proposed to solve the problem of poor classification of KNN algorithms in the environment of small sample pools. In the case where insufficient classed samples are available, the local thematic features of the samples to be classified are amplified by retrieval to obtain similar samples with sufficiently fixed classes. The literature [97] gives parallel algorithms on reconfigurable mesh machines (RMESH) for K nearest neighbors in Euclidean space, among others. Li et al. proposed a two-layer hierarchical combination of text classification [98]. It was demonstrated that the combination of support vector machine and KNN methods could make SVM noise less sensitive and improve the efficiency of KNN methods, showing better classification performance. Shi et al. [99] propose a semi-supervised classification algorithm based on rough set error, using rough set error theory to extract negative case samples. A new classifier is constructed by combining SVM, Rocchio and Naive Bayes to improve the classification accuracy.

## 3. Structure of HHO

HHO is a heuristic algorithm put forward by Heidari et al that was enlightened by the predatory behaviors of Harris's hawks [35]. In line with other heuristic algorithms, HHO contains exploration and exploitation phases. In the two different phases, Harris's hawks produce different predatory behaviors.

### 3.1. Exploration phase

Harris Hawks in the exploration phase roam many locations waiting for prey to arrive. Harris's Hawks usually consider the position of their companions and prey when selecting their perches. Two main strategies are followed during this phase, as shown in Eq (2.1).

$$X(t+1) = \begin{cases} X_{rand}(t) - r_1|X_{rand}(t) - 2r_2X(t)| & q \geq 0.5 \\ (X_{rabbit}(t) - X_m(t)) - r_3(LB + r_4(UB - LB)) & q < 0.5 \end{cases} \qquad \text{Eq (2.1)}$$

$$X_m(t) = \frac{1}{N}\sum_{i=1}^{N} X_i(t) \qquad \text{Eq (2.2)}$$

In Eq (2.1)., $X(t+1)$ is the next updated position of the population iteration of Harris's hawk. $X_{rand}(t)$ refers to a random position in the current population. $X_{rabbit}(t)$ refers to the

prey's location, i.e., the location of the current optimal solution. $r_1$, $r_2$, $r_3$, $r_4$ and $q$ are random numbers between [0,1], respectively. *UB* and *LB* are used to control the upper boundary and lower bound, independently. In Eq (2.2), $X_m(t)$ is the average position of individuals in the current Harris's hawk population. where $N$ is the population size and $X_i(t)$ is the position of the *i*-th individual.

## 3.2. Transition factor

The escape energy E of the target is designed during the exploitation phase, which determines the exploitation behavior of the Harris Hawk, as shown in Eq (2.3).

$$E = 2(1 - t/T) \qquad\qquad \text{Eq (2.3)}$$

$$El = E \times E_0 \qquad\qquad \text{Eq (2.4)}$$

where $E$ represents the escape energy of the target prey. $E_0$ denotes the escape energy of target in its initial state. $T$ represents the maximum number of iterations. $t$ denotes the current number of iterations. Harris Hawk chooses different exploitation behaviors depending on the value of $E$. When $E \geq 1$, the Harris Hawk searches for the target in a region far from the current solution. When $E < 1$, Harris Hawk searches in the neighborhood of the current solution.

## 3.3. Exploitation phase

In the process of the exploitation phase, Harris's Hawks choose the most appropriate way to attack their prey, depending on each situation. Four strategies describe the hunting behaviors of Harris's hawk. The probability of prey escape is set to $r$. $r < 0.5$ represents the state in which the prey escapes capture. $r \geq 0.5$ represents the state in which the prey is captured. Regardless of the state, the Harris hawk always surrounds the prey based on escape energy. When $E \geq 0.5$, the Harris Hawk performs a soft besiege. when $E < 0.5$, the Harris Hawk performs a hard besiege.

**3.3.1 Soft besiege.** When $E \geq 0.5$ and $r \geq 0.5$, then Harris Hawk successfully captures the prey in a soft besiege state. In soft besiege, the Harris Hawk surrounds its prey and dissipates the prey's energy. The process can be expressed as Eq (2.5).

$$X(t + 1) = \Delta X(t) - El|JX_{rand} - X(t)| \qquad\qquad \text{Eq (2.5)}$$

where $\Delta X(t)$ denotes the vector difference between the prey target and the current individual and can be represented as Eq (2.6). $J$ refers to the prey escape force, as shown in Eq (2.7).

$$\Delta X(t) = X_{rabbit}(t) - X(t) \qquad\qquad \text{Eq (2.6)}$$

$$J = 2(1 - r_5) \qquad\qquad \text{Eq (2.7)}$$

where $r_5$ represents a random number between [0,1].

**3.3.2 Hard besiege.** When $E < 0.5$ and $r \geq 0.5$, the energy of the prey target is at a low value. At this point, Harris's hawk uses a surprise attack to approach the prey, a process that can be expressed by Eq (2.8).

$$X(t + 1) = X_{rabbit}(t) - El|\Delta X(t)| \qquad\qquad \text{Eq (2.8)}$$

**3.3.3 Soft besiege with progressive rapid dives.** When $E \geq 0.5$ and $r < 0.5$, the energy of the prey target is at a high value. The probability of prey avoiding capture by Harris's hawk

was high at this time. Harris's hawks performed soft besiege between prey captures. The subsequent random wandering of Levy-flight was used to simulate prey escape and Harris Hawk pursuit, a process that can be expressed by Eq (2.9). Eq (2.10) and Eq (2.11) determine the next updated position of the Harris Hawk.

$$LF(x) = \frac{u \times \sigma}{|v|^{\frac{1}{\beta}}}, \sigma = \left( \frac{\Gamma(1 + \beta) \times sin\left(\frac{\pi\beta}{2}\right)}{\Gamma\left(\frac{1+\beta}{2}\right) \times \beta \times 2\left(\frac{\beta-1}{2}\right)} \right)^{\frac{1}{\beta}} \qquad \text{Eq (2.9)}$$

$$Y = X_{rabbit}(t) - El|JX_{rabbit}(t) - X(t)| \qquad \text{Eq (2.10)}$$

$$Z = Y + S \times LF(D) \qquad \text{Eq (2.11)}$$

where $u$ and $v$ are both random numbers between [0,1], $\beta$ takes 1.5. $D$ is the dimension of the problem. $S$ vector is randomly generated. Greedy selection is then used to determine which positional update is more favorable to the optimal solution, as shown in Eq (2.12). $F()$ calculates the target fitness value.

$$X(t + 1) = \begin{cases} Y & \text{if } F(Y) < F(X(t)) \\ Z & \text{if } F(Z) < F(X(t)) \end{cases} \qquad \text{Eq (2.12)}$$

**3.3.4 Hard besiege with progressive rapid dives.** When $E<0.5$ and $r<0.5$, Harris's Hawk surrounds the low-energy prey target by forming an envelope. In the process, Harris's hawk approaches the prey by narrowing the envelope. This process is described as Eq (2.13)–Eq (2.15).

$$Y' = X_{rabbit}(t) - El|J*X_{rabbit}(t) - X_m(t)| \qquad \text{Eq (2.13)}$$

$$Z' = Y' + S \times LF(D) \qquad \text{Eq (2.14)}$$

$$X(t + 1) = \begin{cases} Y' & \text{if } F(Y') < F(X(t)) \\ Z' & \text{if } F(Z') < F(X(t)) \end{cases} \qquad \text{Eq (2.15)}$$

## 4. Proposed SGHHO

In the subsection, we systematically present the algorithmic implementation of SGHHO, including the two novel mechanisms introduced therein. Compared to the standard HHO, the proposed SGHHO has better performance.

### 4.1. Simulated annealing based HHO

Simulated annealing (SA) algorithm is a random search algorithm first proposed by N. Metropolis et al [100]. Metropolis perceived a strong similarity between the solid annealing processes in the physical world and the general engineering combinatorial optimization situation. SA is a generally probability-based optimization algorithm where the solid annealing process can be summarized in three parts: the heating stage, the isothermal procedure and the cooling process:

1. Heating stage. This process is designed to increase the energy of the molecules of a solid so that the molecules within it have a more intense thermal movement, thus departing from the equilibrium state of the solid molecules. As the temperature reaches a sufficient value, the solid becomes a liquid, resulting in the disappearance of the non-uniform state in the solid system.

2. Isothermal processes. The states in the system always proceed in the direction of decreasing energy, and as the energy decreases, the system eventually enters equilibrium.

3. The cooling processes. The thermal movement of molecules is reduced, the energy of the system decreases, and a crystalline structure is formed.

The main idea of the simulated annealing algorithm in a combinatorial optimization problem is then shown as follows:

- Initialize the temperature values and solve the optimization problem.

- Execute the algorithm according to the Metropolis criterion.

- Output optimal solution.

In the SA algorithm, the Metropolis criterion is the assumption that the position of the previous agent is $X(n)$. After further iterative updates of the population, the agent position is updated to $X(n+1)$. At the same time, the energy of the search agent changes from $E(n)$ to $E(n+1)$. The probability of receiving a search agent from $X(n)$ to $X(n+1)$ is $p$.

$$p = \begin{cases} 1, & , E(n+1) < E(n) \\ \exp\left(-\dfrac{E(n+1) - E(n)}{T}\right), & E(n+1) \geq E(n) \end{cases}$$ 

Eq (3.1)

where $T$ denotes the relative temperature; when moving to the next iteration of the update, if the energy has become smaller, then this change is accepted. If the energy has increased, the search agent has moved further away from the global optimum. In this case, the algorithm does not immediately discard it but judges it by probability: a random number $\varepsilon$ is generated on a fixed interval [0,1]. If $\varepsilon < p$, the transfer in this case will also be accepted, otherwise the transfer fails to proceed to the next step of the cooling process, and so on. The cooling equation for the above cooling process is:

$$T = T \times q$$ 

Eq (3.2)

where $q$ denotes the cooling factor, the value of which is usually set to 0.99 [28]; T denotes the current temperature. When the temperature drops to the termination temperature Tend, the optimal value is output.

In this study, chaos mapping is introduced for the simulated annealing algorithm to further enhance the performance of SA. The operation of chaos mapping is shown in detail as follows:

$$\beta_{i+1} = \mu \beta_i \times (1 - \beta_i), i = 1, 2 \ldots s - 1$$ 

Eq (3.3)

where $\mu$ denotes the control parameter, whose value is usually set to 4 [101]; $\beta_i$ represents a random value ranging in [0,1]; and $s$ stands for the number of the entire population. The chaotic strategy is strongly influenced by the initial conditions and can be understood as a search movement with a combination of overall ergodicity and randomness [102,103]. By effectively circumventing premature convergence of the population, this approach proficiently mitigates the occurrence thereof, expedites the convergence rate, and enhances the precision of solutions attained through the HHO algorithm.

## 4.2. Gaussian bare-bones

Gaussian Bare Bones is widely used as a variational strategy. Wang et al [104] introduced Gaussian skeleton variation into DE by intersecting the feasible solution obtained from Gaussian perturbation with the solution of the original DE. In this study, enhanced Gaussian Bare Bones were designed to be applied to HHO, and the new update procedure is shown in Eq (3.4).

$$X_{i,j}(t+1) = \begin{cases} N(\mu, \sigma), & rand < CR \\ X_{i_1,j}(t) + r_3 * (X_{i_2,j}(t) - X_{i_3,j}(t)), & otherwise \end{cases} \qquad \text{Eq (3.4)}$$

$$\mu = (X_{best,j} + X_{i,j}(t))/2 \qquad \text{Eq (3.5)}$$

$$\sigma = b * |X_{best,j} - X_{i,j}(t)| \qquad \text{Eq (3.6)}$$

where $N(\mu, \sigma)$ is a Gaussian function with mean $\mu$ and variance $\sigma$. CR is the crossover factor, and its value is usually set to 0.3 [104–106]. $X_{i_1,j}, X_{i_2,j}, X_{i_3,j}$ are three random individuals in dimension $j$, while $i \neq i_1 \neq i_2 \neq i_3$.

In the modified Gaussian Bare Bones, the new update position is generated by a Gaussian distribution. The method calculates the average of the intermediate position between the present capture and the optimal solution, utilizing the difference between the current solution and the optimal solution as a measure of variance. During the initial stages of evolution, when the disparity between the current solution and the global optimum is substantial, the population's individuals encompass the entire search space. This stage primarily emphasizes global exploration. As the number of offspring increases, the deviation decreases, while the use of a linear decreasing factor $b$ ensures that individuals in the later population will cover a smaller search area. At this point the algorithm can move more quickly into the behaviors of local exploitation, ensuring convergence at a later stage.

## 4.3. Framework of SGHHO

The entire framework of SGHHO combines both of these strategies and the flow graph of its overall framework is detailed in Fig 1. First, the population needs to be set up initially and the optimal position selected as the rabbit's position in the algorithm. After the SGHHO algorithm is updated using the core formulation of HHO, a simulated annealing strategy is introduced. Guided by the SA strategy, the iterative process of SGHHO can accept points with larger energy values, i.e., poorer solutions. As a result, the strategy possesses a better vertical search capability with a larger search range, thus effectively improving the probability of HHO finding a global optimum. Next, Gaussian bare bones strategy is implemented, and the process of fine-tuning the Gaussian sampling allows SGHHO to ensure diversity exploration in the first stage while focusing on exploitation in the next phase. The switch in search behavior from exploration to exploitation allows SGHHO to converge to a higher level of accuracy.

Eventually, the optimal solution in the entire population is updated until it reaches the maximum allowed number of iterations. Additionally, Algorithm 1 showcases the pseudo-code for the SGHHO method employed in this research.

**Algorithm1. The pseudocode of SGHHO.**

```
Initialize the population size N and maximum number of evaluation
MaxFES;
Randomly initialize the location information of population
Xi(i = 1,2,...,N);
While (termination condition is not satisfied) do
```

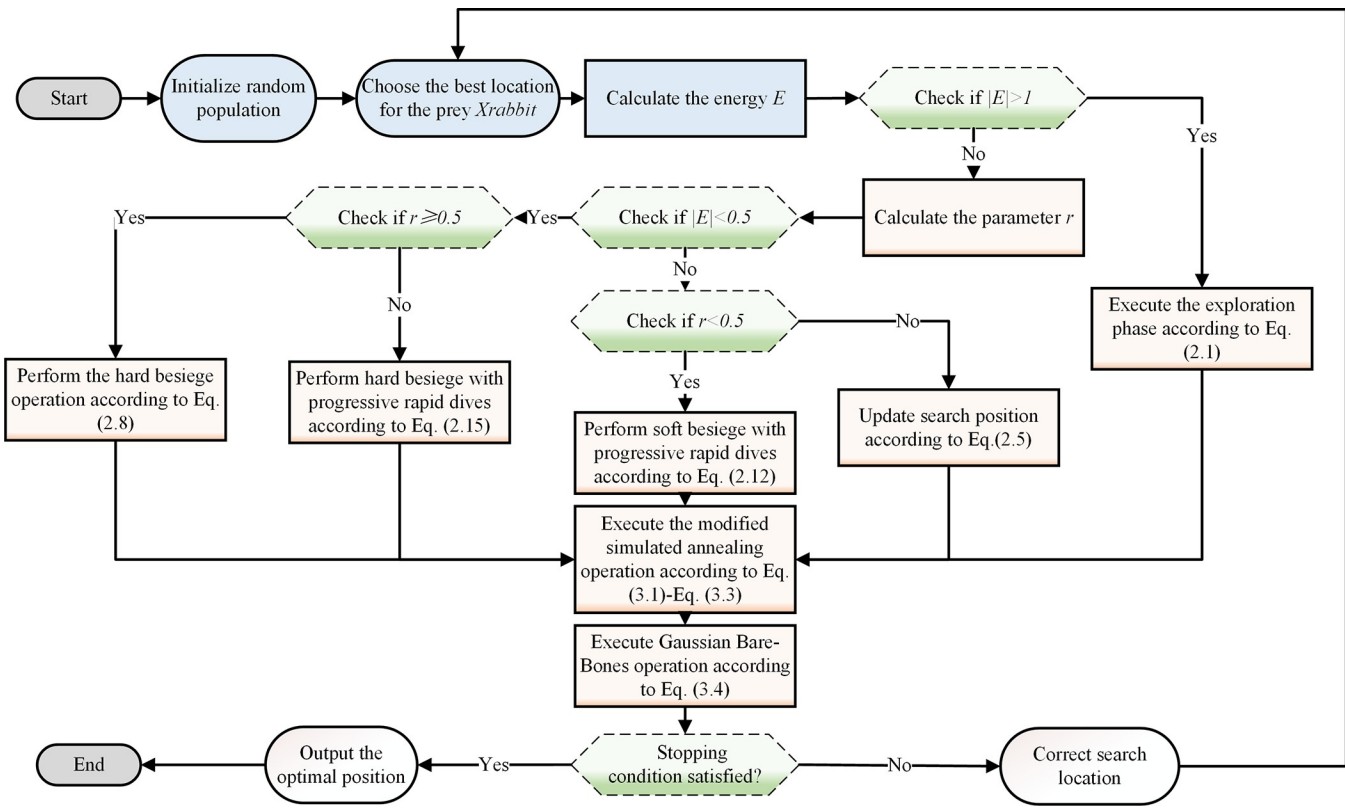

**Fig 1. Flowchart of SGHHO.**

```
Calculate the fitness value for each individual;
Set X_rabbit to the current optimal solution;
for (Each individual) do
    Updating the initial escape energy E₀ and escape force J;
    Update E by Eq (2.3);
    if (|E| ≥ 1) then
            if (q < 0.5) then
                    Updating the position according to Eq (2.1);
            elseif (q ≥ 0.5) then
                    Update the location according to the random
selection in Eq (2.1);
            end
            if (|E| < 1) then
                if (r ≥ 0.5 and |E| ≥ 0.5) then
                    Use Eq (2.5) to update the position;
                elseif (r ≥ 0.5 and |E| < 0.5) then
                    Use Eq (2.8) to update the position;
                elseif (r < 0.5 and |E| ≥ 0.5) then
                    Use Eq (2.12) to update the position;
                elseif (r < 0.5 and |E| < 0.5) then
                    Use Eq (2.15) to update the position;
    end if
        end for
    Calculate the fitness value for all individuals;
    Calculate the mutation position of the modified Gaussian bare
bone strategy by Eq (2.15);
```

```
        Updating search agent locations by means of the modified simu-
lated annealing strategy;
t = t + 1;
end while
return X_{rabbit};
```

## 4.4. Computational complexity

The evaluation of algorithmic models involves considering the time complexity of an optimizer, which is an important metric. In this research, the time complexity of SGHHO consists of various steps, including population initialization, updating fitness values, modifying the positions of individuals in the population, executing the simulated annealing strategy, and implementing the Gaussian bare bone strategy. In the standard SGHHO algorithm, the time complexity of these first three components is calculated as O (population initialization) = O (N); O (fitness value update) = O(T*N) during T update iterations; and O (population individual position update) = O(T*N*D) during T iterations. where N denotes the assumed population size; T denotes the maximum number of iterative updates during the experiment; and D denotes the problem dimensions. The SA strategy and GB strategy are also included in the SGHHO algorithm to help SGHHO improve its performance. In particular, the time complexity of the SA strategy is O (SA strategy) = O(N); the time complexity of GB strategy is O (GB strategy) = O(N*D). So that the overall time complexity consumed by the SGHHO algorithm is O(SGHHO) = O(N*(T+TD+2+D)).

## 5. Proposed SGHHO-KNN

In this section, to enable more representative attributes to be found in the dataset and helps researchers to mine more effective information. In this paper, a simulated annealing mechanism and a Gaussian bare bone variation strategy for improving HHO is proposed for feature selection. Various evaluation criteria exist for different feature selection methods. In this research, a wrapped feature selection method is employed, which primarily relies on the learning algorithm. The classification performance of feature selection serves as the evaluation metric for this method. To enhance the evaluation process, this study incorporates a KNN classifier. KNN is a non-parametric classification technique known for its ability to achieve high accuracy when dealing with unknown and non-normally distributed data. Moreover, KNN offers several advantages, including conceptual clarity and ease of implementation. Furthermore, the KNN approach primarily depends on a small set of neighboring samples in its classification process, rather than relying on class domain discrimination methods. This characteristic makes the KNN method particularly well-suited for datasets with overlapping or intersecting class domains. It is worth noting that the classification error rate of the KNN classifier is closely tied to the concept of distance. In general, KNN algorithms will use distance metrics such as Euclidean and Manhattan for operations. In this study, Manhattan is used for the sample calculation.

In overview, the SGHHO-KNN model is developed in this paper. The model involves three main components: primarily, in this chapter, a refined version of the HHO algorithm is introduced, which incorporates a simulated annealing mechanism and a Gaussian skeleton variation strategy. Subsequently, the SGHHO algorithm is synergistically combined with the K-Nearest Neighbor (KNN) model, giving rise to a novel classification model referred to as SGHHO-KNN. This model utilizes a hybrid feature selection approach. Within this framework, the SGHHO algorithm is employed to conduct a thorough exploration of the feature space, enabling the identification and selection of the most optimal feature subset. The K-Nearest Neighbor classifier is mainly used for feature subset evaluation and for comparing

the classification performance after feature selection in combination with the comparison algorithm. That is, with ten-fold cross-validation, the SGHHO algorithm finds the best feature subset on the training set by internal five-fold cross-validation, and the KNN model uses this best feature subset to perform the classification task on the test set. In addition, for the experiments in this study, each classification experiment was run 10 times independently to ensure that the algorithm was fair and unbiased. Therefore, the performance of the algorithm was evaluated according to the average classification results and the corresponding standard deviation of each of the 10 independent runs. The flow chart of the SGHHO-KNN hybrid model proposed in this study is shown in Fig 2.

## 6. Experimental design and analysis of results

To analyze the SGHHO's performance in various aspects, comprehensive experiments are designed. In the optimization experiments, comprehensive experiments on the IEEE CEC 2014 function set had been conducted. The experimental design procedure of the work is shown as follows: firstly, experiments are designed to test the scalability of the SGHHO algorithm for dimensional changes; secondly, the effects of two mechanisms on the SGHHO algorithm are investigated; thirdly, comparison experiments are carried out between the SGHHO algorithm and 10 well-known optimization algorithms. In addition, for the metaheuristic algorithm setup described above, the population size is 30; the maximum number of evaluations is 300,000. Meanwhile, all competing methods were independently repeated 30 times during the testing process to avoid chance during experiments. Finally, all the above tests were conducted on Windows 10 host with 16GB RAM and 3.6GHz main frequency and coded using MATLAB R2016b.

### 6.1. CEC 2014 function validation

In the section, IEEE CEC 2014 function set was chosen to prove each algorithm's performance in optimization precision and convergence speed. In particular, algorithms are divided into four main types: singlet functions ($F_1$- $F_3$); multimodal functions (F4—$F_8$); hybrid modal functions ($F_9$- $F_{20}$) and composite modal functions ($F_{21}$- $F_{30}$). Table 1 shows the specific data for these 30 functions. In the table, the last two columns show the type of each algorithm and the corresponding global optimum value. Furthermore, to provide a more comprehensive assessment of each algorithm's performance, two metrics, namely the average value (AVG) and the standard deviation value (STD), are utilized. The experimental tables highlight the optimal values for each problem in bold black font, ensuring their prominence. Additionally, non-parametric statistical tests such as the Wilcoxon signed-rank test [107] and the Friedman test [108] are employed to quantitatively evaluate the performance of SGHHO.

### 6.2. Scalability analysis of SGHHO

The problem's dimensionality corresponds to the number of factors to be optimized within the given optimization problem. Consequently, optimizing high-dimensional data allows for a more comprehensive evaluation of the algorithm's overall performance and stability. Thus, the performance of the SGHHO algorithm can be thoroughly assessed. We use the original HHO and Farmland Fertility Algorithm (FFA) [109], which have better performance in the variable dimensionality case, as the reference objects for the experiments in this subsection. Within the experimental framework, scalability experiments are conducted using a test function set consisting of 30 functions from IEEE CEC 2014. The dimensions are varied, specifically set to 100, 500, and 1000, respectively. All experimental parameters remain constant, with the exception of the dimension setting (D). The metrics utilized to analyze the experimental results include

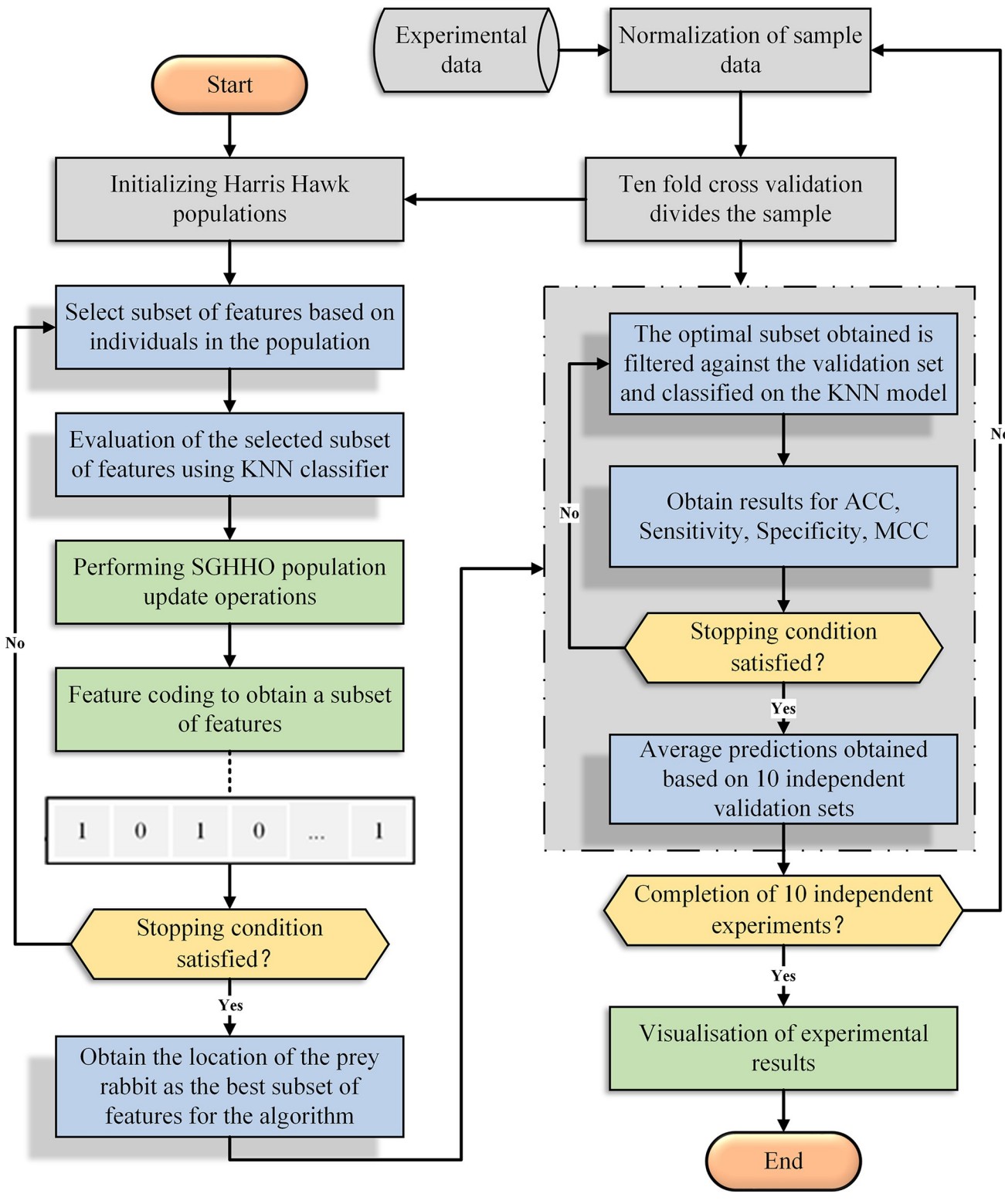

**Fig 2. Framework structure of SGHHO-KNN.**

**Table 1. Description of the 30 IEEE CEC 2014 benchmark functions.**

| ID | Function | Class | Optimum |
|---|---|---|---|
| **F1** | Rotated High Conditioned Elliptic Function | Unimodal | 100 |
| **F2** | Rotated Bent Cigar Function | Unimodal | 200 |
| **F3** | Rotated Discus Function | Unimodal | 300 |
| **F4** | Shifted and Rotated Rosenbrock's Function | Multimodal | 400 |
| **F5** | Shifted and Rotated Ackley's Function | Multimodal | 500 |
| **F6** | Shifted and Rotated Weierstrass Function | Multimodal | 600 |
| **F7** | Shifted and Rotated Griewank's Function | Multimodal | 700 |
| **F8** | Shifted Rastrigin's Function | Multimodal | 800 |
| **F9** | Shifted and Rotated Rastrigin's Function | Hybrid | 900 |
| **F10** | Shifted Schwefel's Function | Hybrid | 1000 |
| **F11** | Shifted and Rotated Schwefel's Function | Hybrid | 1100 |
| **F12** | Shifted and Rotated Katsuura Function | Hybrid | 1200 |
| **F13** | Shifted and Rotated HappyCat Function | Hybrid | 1300 |
| **F14** | Shifted and Rotated HGBat Function | Hybrid | 1400 |
| **F15** | Shifted and Rotated Expanded Griewank's plus Rosenbrock's Function | Hybrid | 1500 |
| **F16** | Shifted and Rotated Expanded Scaffer's F6 Function | Hybrid | 1600 |
| **F17** | Hybrid Function 1 (N = 3) | Hybrid | 1700 |
| **F18** | Hybrid Function 2 (N = 3) | Hybrid | 1800 |
| **F19** | Hybrid Function 3 (N = 4) | Hybrid | 1900 |
| **F20** | Hybrid Function 4 (N = 4) | Hybrid | 2000 |
| **F21** | Hybrid Function 5 (N = 5) | Composition | 2100 |
| **F22** | Hybrid Function 6 (N = 5) | Composition | 2200 |
| **F23** | Composition Function 1 (N = 5) | Composition | 2300 |
| **F24** | Composition Function 2 (N = 3) | Composition | 2400 |
| **F25** | Composition Function 3 (N = 3) | Composition | 2500 |
| **F26** | Composition Function 4 (N = 5) | Composition | 2600 |
| **F27** | Composition Function 5 (N = 5) | Composition | 2700 |
| **F28** | Composition Function 6 (N = 5) | Composition | 2800 |
| **F29** | Composition Function 7 (N = 3) | Composition | 2900 |
| **F30** | Composition Function 8 (N = 3) | Composition | 3000 |

AVG (average) and STD (standard deviation). A comprehensive overview of the results can be found in Table 2.

The table provides a clear visual representation, demonstrating that the SGHHO algorithm consistently outperforms the original HHO algorithm in unimodal functions (F1-F3), irrespective of the dimensional complexity. The AVG values achieved by the SGHHO algorithm exhibit a significant improvement compared to those of the original HHO algorithm. This superiority is evident in both low and high-dimensional problem settings. Also, SGHHO outperforms the FFA algorithm for F1 and F3 functions. And with the increasing complexity of the functions, the SGHHO algorithm has better optimization performance compared with the original HHO algorithm in multimodal functions (F4-F9). The implementation of the GB strategy successfully directs the HHO algorithm towards exploring regions that contain more optimal solutions, thereby enhancing the algorithm's precision in generating solutions. The FFA, with its excessive local search capability, slightly outperforms the SGHHO on F4, F7 and F8. Further, in the mixed modal functions (F9-F20), the SGHHO algorithm still maintains its obvious superiority over FFA and HHO. Despite the variations in dimensional settings, the SGHHO algorithm demonstrates consistent performance in attaining optimal solutions.

**Table 2. Experimental results of scalability tests on SGHHO.**

| F | Metric | 100 | | | 500 | | | 1000 | | |
|---|---|---|---|---|---|---|---|---|---|---|
| | | SGHHO | FFA | HHO | SGHHO | FFA | HHO | SGHHO | FFA | HHO |
| F1 | AVG | **1.64E+06** | 9.44E+06 | 1.38E+07 | **1.37E+06** | 9.49E+06 | 1.43E+07 | **1.17E+06** | 9.56E+06 | 1.20E+07 |
| | STD | **9.50E+05** | 4.34E+06 | 5.59E+06 | **3.93E+05** | 3.90E+06 | 6.71E+06 | **5.50E+05** | 4.13E+06 | 4.87E+06 |
| F2 | AVG | 1.66E+04 | **5.86E+02** | 1.46E+07 | 2.38E+04 | **3.80E+02** | 1.38E+07 | 1.61E+04 | **7.96E+02** | 1.42E+07 |
| | STD | 9.36E+03 | **1.47E+03** | 3.15E+06 | 3.43E+04 | **6.07E+02** | 3.05E+06 | 1.11E+04 | **1.66E+03** | 2.87E+06 |
| F3 | AVG | **3.60E+02** | 1.91E+03 | 1.03E+04 | **3.49E+02** | 1.70E+03 | 9.48E+03 | **3.45E+02** | 1.59E+03 | 1.07E+04 |
| | STD | **3.57E+01** | 1.29E+03 | 4.51E+03 | **3.56E+01** | 9.20E+02 | 3.44E+03 | **2.18E+01** | 7.38E+02 | 3.48E+03 |
| F4 | AVG | 4.69E+02 | **4.60E+02** | 5.58E+02 | 4.80E+02 | **4.63E+02** | 5.55E+02 | 4.70E+02 | **4.47E+02** | 5.68E+02 |
| | STD | 3.49E+01 | **3.44E+01** | 7.08E+01 | 3.87E+01 | **3.75E+01** | 4.35E+01 | 2.98E+01 | **3.64E+01** | 5.26E+01 |
| F5 | AVG | **5.20E+02** | 5.21E+02 | 5.20E+02 | **5.20E+02** | 5.21E+02 | 5.20E+02 | **5.20E+02** | 5.21E+02 | 5.20E+02 |
| | STD | 1.36E-01 | 4.97E-02 | 1.61E-01 | 4.80E-02 | 6.56E-02 | 1.85E-01 | 1.12E-01 | 5.95E-02 | 1.59E-01 |
| F6 | AVG | 6.21E+02 | **6.18E+02** | 6.32E+02 | 6.23E+02 | **6.17E+02** | 6.30E+02 | 6.22E+02 | **6.18E+02** | 6.31E+02 |
| | STD | 3.19E+00 | 4.62E+00 | **2.99E+00** | 2.92E+00 | 6.41E+00 | 4.51E+00 | 3.42E+00 | 4.60E+00 | 3.66E+00 |
| F7 | AVG | **7.00E+02** | 7.00E+02 | 7.01E+02 | **7.00E+02** | 7.00E+02 | 7.01E+02 | **7.00E+02** | 7.00E+02 | 7.01E+02 |
| | STD | 3.48E-02 | **1.88E-03** | 3.14E-02 | 2.80E-02 | **6.43E-03** | 2.48E-02 | 3.71E-02 | **2.57E-03** | 3.03E-02 |
| F8 | AVG | 8.42E+02 | **8.19E+02** | 9.07E+02 | 8.41E+02 | **8.18E+02** | 9.14E+02 | 8.41E+02 | **8.20E+02** | 9.13E+02 |
| | STD | 9.06E+00 | **4.90E+00** | 1.57E+01 | 7.32E+00 | **4.68E+00** | 1.57E+01 | 7.81E+00 | **5.99E+00** | 1.51E+01 |
| F9 | AVG | 1.08E+03 | **1.05E+03** | 1.09E+03 | 1.08E+03 | **1.06E+03** | 1.09E+03 | 1.07E+03 | **1.05E+03** | 1.09E+03 |
| | STD | **1.64E+01** | 2.11E+01 | 2.01E+01 | **2.16E+01** | 2.34E+01 | 2.36E+01 | **1.94E+01** | 3.06E+01 | 2.12E+01 |
| F10 | AVG | 1.62E+03 | **1.50E+03** | 3.16E+03 | **1.48E+03** | 1.54E+03 | 3.05E+03 | **1.40E+03** | 1.48E+03 | 3.28E+03 |
| | STD | 3.06E+02 | 5.71E+02 | 7.77E+02 | **2.71E+02** | 4.31E+02 | 7.57E+02 | 3.46E+02 | **3.05E+02** | 8.94E+02 |
| F11 | AVG | **4.45E+03** | 7.60E+03 | 5.27E+03 | **4.55E+03** | 7.57E+03 | 4.98E+03 | **4.37E+03** | 7.59E+03 | 5.33E+03 |
| | STD | **4.68E+02** | 4.56E+02 | 7.26E+02 | 5.97E+02 | **3.61E+02** | 9.43E+02 | 5.17E+02 | **4.72E+02** | 6.60E+02 |
| F12 | AVG | **1.20E+03** | 1.20E+03 | 1.20E+03 | **1.20E+03** | 1.20E+03 | 1.20E+03 | **1.20E+03** | 1.20E+03 | 1.20E+03 |
| | STD | **2.23E-01** | 3.00E-01 | 4.19E-01 | **2.14E-01** | 2.73E-01 | 4.70E-01 | **1.79E-01** | 2.66E-01 | 4.99E-01 |
| F13 | AVG | **1.30E+03** | 1.30E+03 | 1.30E+03 | **1.30E+03** | 1.30E+03 | 1.30E+03 | **1.30E+03** | 1.30E+03 | 1.30E+03 |
| | STD | 8.92E-02 | 5.02E-02 | 1.40E-01 | 7.31E-02 | 4.73E-02 | 1.26E-01 | 9.16E-02 | 4.37E-02 | 1.09E-01 |
| F14 | AVG | **1.40E+03** | 1.40E+03 | 1.40E+03 | **1.40E+03** | 1.40E+03 | 1.40E+03 | **1.40E+03** | 1.40E+03 | 1.40E+03 |
| | STD | **3.54E-02** | 3.75E-02 | 1.35E-01 | 3.74E-02 | 3.22E-02 | 5.16E-02 | 4.14E-02 | 3.86E-02 | 4.70E-02 |
| F15 | AVG | **1.51E+03** | 1.52E+03 | 1.54E+03 | **1.51E+03** | 1.52E+03 | 1.54E+03 | **1.51E+03** | 1.52E+03 | 1.54E+03 |
| | STD | 2.14E+00 | 1.20E+00 | 9.79E+00 | 2.82E+00 | 1.10E+00 | 9.13E+00 | 2.48E+00 | 1.13E+00 | 7.33E+00 |
| F16 | AVG | **1.61E+03** | 1.61E+03 | 1.61E+03 | **1.61E+03** | 1.61E+03 | 1.61E+03 | **1.61E+03** | 1.61E+03 | 1.61E+03 |
| | STD | 3.54E-01 | 2.10E-01 | 3.69E-01 | **2.68E-01** | 2.39E-01 | 4.13E-01 | **2.99E-01** | 2.79E-01 | 3.53E-01 |
| F17 | AVG | **1.26E+05** | 4.95E+05 | 2.00E+06 | **1.40E+05** | 4.52E+05 | 1.79E+06 | **1.67E+05** | 4.42E+05 | 2.04E+06 |
| | STD | **6.40E+04** | 2.74E+05 | 1.36E+06 | **5.57E+04** | 3.06E+05 | 1.04E+06 | **1.44E+05** | 3.05E+05 | 1.65E+06 |
| F18 | AVG | 4.02E+03 | **3.30E+03** | 1.07E+05 | 4.25E+03 | **3.23E+03** | 1.50E+05 | 4.29E+03 | **3.01E+03** | 1.25E+05 |
| | STD | 4.12E+03 | **2.07E+03** | 3.98E+04 | 5.27E+03 | **1.64E+03** | 2.43E+05 | 4.82E+03 | **1.65E+03** | 7.01E+04 |
| F19 | AVG | 1.92E+03 | **1.91E+03** | 1.94E+03 | 1.92E+03 | **1.91E+03** | 1.93E+03 | 1.92E+03 | **1.91E+03** | 1.93E+03 |
| | STD | 2.67E+00 | **1.22E+00** | 4.31E+01 | 2.84E+00 | **1.06E+00** | 2.58E+01 | 3.02E+00 | **1.28E+00** | 2.79E+01 |
| F20 | AVG | **2.30E+03** | 3.84E+03 | 1.58E+04 | **2.27E+03** | 3.97E+03 | 1.74E+04 | **2.30E+03** | 3.86E+03 | 1.66E+04 |
| | STD | **6.64E+01** | 7.95E+02 | 5.99E+03 | **7.00E+01** | 1.35E+03 | 8.56E+03 | **6.91E+01** | 1.06E+03 | 6.07E+03 |
| F21 | AVG | **5.81E+04** | 1.45E+05 | 5.89E+05 | **6.76E+04** | 1.33E+05 | 6.04E+05 | **6.29E+04** | 1.29E+05 | 6.53E+05 |
| | STD | **3.28E+04** | 9.19E+04 | 6.61E+05 | **3.41E+04** | 7.23E+04 | 4.69E+05 | **4.07E+04** | 9.05E+04 | 5.83E+05 |
| F22 | AVG | 2.72E+03 | **2.39E+03** | 3.03E+03 | 2.74E+03 | **2.43E+03** | 3.09E+03 | 2.70E+03 | **2.44E+03** | 3.10E+03 |
| | STD | 1.60E+02 | **9.77E+01** | 1.92E+02 | 1.70E+02 | **1.02E+02** | 2.65E+02 | 1.30E+02 | **1.13E+02** | 3.15E+02 |
| F23 | AVG | **2.50E+03** | 2.62E+03 | 2.50E+03 | **2.50E+03** | 2.62E+03 | 2.50E+03 | **2.50E+03** | 2.62E+03 | 2.50E+03 |
| | STD | **0.00E+00** | 1.39E-12 | 0.00E+00 | **0.00E+00** | 1.24E-12 | 0.00E+00 | **0.00E+00** | 1.49E-12 | 0.00E+00 |

*(Continued)*

**Table 2.** (Continued)

| F | Metric | 100 | | | 500 | | | 1000 | | |
|---|---|---|---|---|---|---|---|---|---|---|
| | | SGHHO | FFA | HHO | SGHHO | FFA | HHO | SGHHO | FFA | HHO |
| F24 | AVG | **2.60E+03** | 2.62E+03 | **2.60E+03** | **2.60E+03** | 2.63E+03 | **2.60E+03** | **2.60E+03** | 2.63E+03 | **2.60E+03** |
| | STD | 2.47E-03 | 1.15E+00 | **4.23E-04** | 7.39E-03 | 1.59E+00 | **7.95E-05** | 3.29E-03 | 2.71E+00 | **4.42E-04** |
| F25 | AVG | **2.70E+03** | 2.71E+03 | 2.70E+03 | **2.70E+03** | 2.71E+03 | 2.70E+03 | **2.70E+03** | 2.71E+03 | 2.70E+03 |
| | STD | **0.00E+00** | 1.90E+00 | 0.00E+00 | **0.00E+00** | 1.45E+00 | 0.00E+00 | **0.00E+00** | 1.88E+00 | 0.00E+00 |
| F26 | AVG | **2.70E+03** | 2.70E+03 | 2.78E+03 | 2.71E+03 | **2.70E+03** | 2.76E+03 | **2.70E+03** | **2.70E+03** | 2.76E+03 |
| | STD | 1.82E+01 | 6.68E-02 | 4.05E+01 | 3.04E+01 | **5.46E-02** | 4.96E+01 | 1.38E-01 | **5.45E-02** | 4.88E+01 |
| F27 | AVG | **2.90E+03** | 3.12E+03 | 2.90E+03 | **2.90E+03** | 3.12E+03 | 2.90E+03 | **2.90E+03** | 3.12E+03 | 2.90E+03 |
| | STD | **0.00E+00** | 1.20E+01 | 0.00E+00 | **0.00E+00** | 4.04E+01 | 0.00E+00 | **0.00E+00** | 3.42E+01 | 0.00E+00 |
| F28 | AVG | **3.00E+03** | 3.65E+03 | 3.00E+03 | **3.00E+03** | 3.66E+03 | 3.00E+03 | **3.00E+03** | 3.67E+03 | 3.00E+03 |
| | STD | **0.00E+00** | 5.40E+01 | 0.00E+00 | **0.00E+00** | 3.10E+01 | 0.00E+00 | **0.00E+00** | 4.45E+01 | 0.00E+00 |
| F29 | AVG | 3.11E+03 | 4.53E+03 | **3.10E+03** | 3.11E+03 | 4.51E+03 | **3.10E+03** | 3.11E+03 | 4.48E+03 | **3.10E+03** |
| | STD | 3.31E+00 | 4.52E+02 | **0.00E+00** | 1.47E+01 | 4.64E+02 | **0.00E+00** | 3.88E+00 | 4.84E+02 | **0.00E+00** |
| F30 | AVG | **3.39E+03** | 4.78E+03 | 6.87E+03 | **3.44E+03** | 4.89E+03 | 7.90E+03 | **3.48E+03** | 4.88E+03 | 6.66E+03 |
| | STD | **2.20E+02** | 3.75E+02 | 9.55E+03 | **3.01E+02** | 7.41E+02 | 7.81E+03 | **5.36E+02** | 6.23E+02 | 6.08E+03 |

Among the composite functions (F21-F30), although there is a slight difference in the AVG value between the SGHHO algorithm and the original algorithm on the F29 function, it remains relatively small. Conversely, for all other functions, the SGHHO algorithm outperforms the HHO algorithm by a significant margin. The results of scalability experiments indicate that the SGHHO algorithm exhibits notable enhancements in optimization performance and stability across 93.3% of the functions in the IEEE CEC 2014 function set when considering variable dimensionality scenarios. This signifies the considerable improvements achieved by the SGHHO algorithm through the integration of SA and GB strategies.

## 6.3. The impact of two mechanisms

This subsection investigates the effect of the randomly introduced simulated annealing strategy and the Gaussian bare bone strategy on HHO. To enable effective verification of the role of the added mechanisms, this experiment performs all sequential sequencing of the SA and GB strategies, thus avoiding interactions between the mechanisms. The details are displayed in Table 3, where "S" and "G" denote "simulated annealing" and "Gaussian skeleton", respectively. 1 and 0 indicate that the strategy is selected and unselected, respectively. The four algorithms were compared on 30 test functions. Table 4 details the Avg and STD values obtained by the algorithms during the experiments. Similarly, the optimal values obtained under each function are marked in bold. Analysis of the table shows that SGHHO has the best solution accuracy out of the 30 functions and ranks first out of 19 functions. In addition, SGHHO is more stable than SHHO and GHHO in terms of STD values. This means that the SGHHO algorithm maximizes the performance benefits of both strategies.

**Table 3. Experimental design of mechanism combinations.**

| | SA | GB |
|---|---|---|
| HHO | 0 | 0 |
| GHHO | 0 | 1 |
| SHHO | 1 | 0 |
| SGHHO | 1 | 1 |

**Table 4. Optimization results of each HHO variants on IEEE CEC 2014 functions.**

| | $F_1$ | | $F_2$ | | $F_3$ | |
|---|---|---|---|---|---|---|
| | AVG | STD | AVG | STD | AVG | STD |
| **SGHHO** | **1.3004E+06** | **5.6615E+05** | 1.4639E+04 | 7.6388E+03 | 3.4932E+02 | 2.4514E+01 |
| GHHO | 1.2163E+07 | 1.2565E+07 | **7.5179E+03** | **2.4676E+04** | **3.1166E+02** | **2.3358E+01** |
| SHHO | 1.8151E+06 | 6.8871E+05 | 3.0546E+06 | 8.5831E+05 | 5.3915E+02 | 1.0844E+02 |
| HHO | 1.6147E+07 | 7.2324E+06 | 1.4291E+07 | 2.6541E+06 | 8.9513E+03 | 2.8344E+03 |

| | $F_4$ | | $F_5$ | | $F_6$ | |
|---|---|---|---|---|---|---|
| | AVG | STD | AVG | STD | AVG | STD |
| **SGHHO** | **4.7760E+02** | **3.7443E+01** | 5.2005E+02 | 7.3211E-02 | 6.2164E+02 | 4.2301E+00 |
| GHHO | 4.9781E+02 | 5.0933E+01 | **5.2003E+02** | **1.2102E-01** | 6.2368E+02 | 3.2191E+00 |
| SHHO | 4.8327E+02 | 4.3772E+01 | 5.2080E+02 | 9.1891E-02 | **6.2115E+02** | **3.1850E+00** |
| HHO | 5.5426E+02 | 4.3131E+01 | 5.2037E+02 | 1.5976E-01 | 6.3095E+02 | 4.1016E+00 |

| | $F_7$ | | $F_8$ | | $F_9$ | |
|---|---|---|---|---|---|---|
| | AVG | STD | AVG | STD | AVG | STD |
| **SGHHO** | 7.0005E+02 | 4.2950E-02 | **8.3916E+02** | **6.1689E+00** | 1.0734E+03 | 1.6391E+01 |
| GHHO | **7.0002E+02** | **2.6385E-02** | 8.6559E+02 | 1.4594E+01 | **1.0717E+03** | **2.6727E+01** |
| SHHO | 7.0106E+02 | 1.4102E-02 | 8.4128E+02 | 8.2343E+00 | 1.0778E+03 | 1.4698E+01 |
| HHO | 7.0112E+02 | 2.8802E-02 | 9.1381E+02 | 1.2948E+01 | 1.0917E+03 | 2.3763E+01 |

| | $F_{10}$ | | $F_{11}$ | | $F_{12}$ | |
|---|---|---|---|---|---|---|
| | AVG | STD | AVG | STD | AVG | STD |
| **SGHHO** | **1.4347E+03** | **3.6411E+02** | 4.5081E+03 | 5.5811E+02 | **1.2005E+03** | **1.7728E-01** |
| GHHO | 1.9418E+03 | 5.2307E+02 | 4.7877E+03 | 6.1852E+02 | 1.2005E+03 | 2.2702E-01 |
| SHHO | 1.4634E+03 | 2.7151E+02 | **4.4575E+03** | **4.7779E+02** | 1.2007E+03 | 2.1117E-01 |
| HHO | 3.3418E+03 | 6.9532E+02 | 5.4497E+03 | 7.2502E+02 | 1.2017E+03 | 4.3400E-01 |

| | $F_{13}$ | | $F_{14}$ | | $F_{15}$ | |
|---|---|---|---|---|---|---|
| | AVG | STD | AVG | STD | AVG | STD |
| **SGHHO** | 1.3004E+03 | 1.0094E-01 | **1.4002E+03** | **4.3851E-02** | **1.5113E+03** | **3.7681E+00** |
| GHHO | 1.3005E+03 | 1.1669E-01 | 1.4003E+03 | 1.5358E-01 | 1.5229E+03 | 8.9626E+00 |
| SHHO | **1.3003E+03** | **6.3199E-02** | 1.4002E+03 | 4.3706E-02 | 1.5149E+03 | 3.3449E+00 |
| HHO | 1.3005E+03 | 1.3686E-01 | 1.4002E+03 | 4.1103E-02 | 1.5379E+03 | 8.0988E+00 |

| | $F_{16}$ | | $F_{17}$ | | $F_{18}$ | |
|---|---|---|---|---|---|---|
| | AVG | STD | AVG | STD | AVG | STD |
| **SGHHO** | 1.6118E+03 | 4.3066E-01 | **1.3856E+05** | **8.5489E+04** | **3.5586E+03** | **2.0875E+03** |
| GHHO | **1.6114E+03** | **6.4796E-01** | 1.2043E+06 | 1.3644E+06 | 7.4275E+03 | 5.6113E+03 |
| SHHO | 1.6119E+03 | 4.2748E-01 | 1.4767E+05 | 6.3563E+04 | 4.5221E+03 | 1.9100E+03 |
| HHO | 1.6124E+03 | 3.7341E-01 | 1.6351E+06 | 1.1048E+06 | 1.7217E+05 | 2.4618E+05 |

| | $F_{19}$ | | $F_{20}$ | | $F_{21}$ | |
|---|---|---|---|---|---|---|
| | AVG | STD | AVG | STD | AVG | STD |
| **SGHHO** | **1.9156E+03** | **3.5495E+00** | **2.2927E+03** | **5.6645E+01** | 5.3399E+04 | **3.3235E+04** |
| GHHO | 1.9192E+03 | 1.3792E+01 | 2.4382E+03 | 3.8903E+02 | 3.3421E+05 | 5.1049E+05 |
| SHHO | 1.9173E+03 | 1.1870E+01 | 2.3228E+03 | 8.1051E+01 | 6.9415E+04 | 3.6753E+04 |
| HHO | 1.9379E+03 | 4.5628E+01 | 1.7912E+04 | 7.5829E+03 | 5.8874E+05 | 3.8244E+05 |

| | $F_{22}$ | | $F_{23}$ | | $F_{24}$ | |
|---|---|---|---|---|---|---|
| | AVG | STD | AVG | STD | AVG | STD |
| **SGHHO** | 2.7440E+03 | 1.7939E+02 | **2.5000E+03** | **0.0000E+00** | **2.6000E+03** | **1.9498E-03** |
| GHHO | **2.7332E+03** | **1.7532E+02** | 2.5116E+03 | 3.5476E+01 | 2.6000E+03 | 1.6949E-04 |
| SHHO | 2.7791E+03 | 1.6018E+02 | 2.5000E+03 | 0.0000E+00 | 2.6000E+03 | 1.7597E-03 |
| HHO | 3.0371E+03 | 2.7336E+02 | 2.5000E+03 | 0.0000E+00 | 2.6000E+03 | 1.7617E-04 |

*(Continued)*

**Table 4.** (Continued)

| $F_{22}$ | $F_{25}$ | | $F_{26}$ | | $F_{27}$ | |
|---|---|---|---|---|---|---|
| | AVG | STD | AVG | STD | AVG | STD |
| **SGHHO** | **2.7000E+03** | **0.0000E+00** | **2.7004E+03** | **6.8531E-02** | **2.9000E+03** | **0.0000E+00** |
| GHHO | **2.7000E+03** | **0.0000E+00** | 2.7072E+03 | 2.5231E+01 | 3.5058E+03 | 2.8702E+02 |
| SHHO | **2.7000E+03** | **0.0000E+00** | 2.7004E+03 | 8.3865E-02 | **2.9000E+03** | **0.0000E+00** |
| HHO | **2.7000E+03** | **0.0000E+00** | 2.7768E+03 | 4.2799E+01 | **2.9000E+03** | **0.0000E+00** |
| | $F_{28}$ | | $F_{29}$ | | $F_{30}$ | |
| | AVG | STD | AVG | STD | AVG | STD |
| **SGHHO** | **3.0000E+03** | **0.0000E+00** | 3.1130E+03 | 1.3423E+01 | **3.3504E+03** | **2.8724E+02** |
| GHHO | 3.2528E+03 | 6.4180E+01 | **3.1081E+03** | **3.3939E+00** | 3.7347E+03 | 3.4315E+02 |
| SHHO | 3.0000E+03 | 0.0000E+00 | 3.1214E+03 | 3.4282E+01 | 3.5270E+03 | 3.7261E+02 |
| HHO | 3.0000E+03 | 0.0000E+00 | 3.7090E+03 | 3.3357E+03 | 8.9845E+03 | 1.1111E+04 |

The p-values for the Wilcoxon signed rank test at a 5% confidence level are presented in Table 5. The symbols "+/-/ = " are utilized to signify the outcomes of the comparison between SGHHO and the other methods. From the table, SGHHO outperforms HHO on 24 test functions; it outperforms the GHHO algorithm on 13 functions; and it outperforms the SHHO algorithm on 10 functions. The integration of the two strategies demonstrates a substantial enhancement in the performance of HHO, attaining its maximum potential in the SGHHO algorithm. Additionally, the Friedman test was employed to quantitatively assess the algorithm's performance. According to the average ranking of the algorithms, the SGHHO algorithm secures the top position among all combinations, signifying a significant advancement in achieving optimal performance. Not only is there a significant improvement in global search, but there is also a significant breakthrough in local exploitation. Therefore, the experimental results in this section serve as evidence for the efficacy of the SA and GB strategies in enhancing the performance of the original HHO algorithm.

## 6.4. Comparison with other reported well-known optimizers

To verify the optimization performance of the SGHHO algorithm, 10 high-quality algorithms including BMWOA [110], CBA [111], EM [112], OBSCA [113], IGWO [114], MFO [115], ALPSO [116], CGPSO [117], SCADE [118] and HGWO [119] were selected for comparison. Furthermore, the IEEE CEC 2014 test set was utilized in the experiments to comprehensively evaluate the algorithms' exploration and exploitation capabilities. To provide further insight into the performance comparisons, statistical tests such as the Wilcoxon rank sum test and the Friedman test were employed to assess the significance differences between SGHHO and the other algorithms. Furthermore, to allow experimental fairness, the whole algorithms were compared under the same settings, as well as each algorithm's parameters were shown in Table 6.

Table 7 shows AVG and STD values gained for all competitors on the set of functions tested. Where the optimal values obtained on each set of functions are bolded in order to allow more visual analysis of the gaps between the algorithms. The results in Table 7 demonstrate that SGHHO can reach the optimal solution of the function on 67% of the tested functions. In particular, on the unimodal function, SGHHO is only not optimal on the F2 function, second only to the ALSPSO and CBA algorithms. Compared to algorithms such as BMWOA, OBSCA, CGPSO and SCADE, the SGHHO showed strong competitiveness on the single-peak function. In addition, although the AVG values of SGHHO on the multimodal functions are not fully

**Table 5. The p-values of Wilcoxon test for HHO variants on the benchmark function.**

| Function | SGHHO | GHHO | SHHO | HHO |
|---|---|---|---|---|
| $F_1$ | N/A | 2.1630E-05 | 1.1138E-03 | 1.7344E-06 |
| $F_2$ | N/A | 6.1564E-04 | 1.7344E-06 | 1.7344E-06 |
| $F_3$ | N/A | 1.9729E-05 | 1.7344E-06 | 1.7344E-06 |
| $F_4$ | N/A | 7.1903E-02 | 4.1653E-01 | 1.0246E-05 |
| $F_5$ | N/A | 6.1564E-04 | 1.7344E-06 | 1.7344E-06 |
| $F_6$ | N/A | 6.2683E-02 | 4.4052E-01 | 2.8786E-06 |
| $F_7$ | N/A | 4.5336E-04 | 1.7344E-06 | 1.7344E-06 |
| $F_8$ | N/A | 2.3534E-06 | 2.6230E-01 | 1.7344E-06 |
| $F_9$ | N/A | 7.4987E-01 | 3.0861E-01 | 3.8542E-03 |
| $F_{10}$ | N/A | 1.4839E-03 | 6.8836E-01 | 1.7344E-06 |
| $F_{11}$ | N/A | 1.5886E-01 | 6.7328E-01 | 4.4493E-05 |
| $F_{12}$ | N/A | 2.2102E-01 | 2.6134E-04 | 1.7344E-06 |
| $F_{13}$ | N/A | 2.8434E-05 | 7.9710E-01 | 2.4118E-04 |
| $F_{14}$ | N/A | 1.7344E-06 | 1.9152E-01 | 1.8910E-04 |
| $F_{15}$ | N/A | 3.1817E-06 | 2.1053E-03 | 1.7344E-06 |
| $F_{16}$ | N/A | 2.5637E-02 | 3.0861E-01 | 1.0246E-05 |
| $F_{17}$ | N/A | 3.8822E-06 | 4.2843E-01 | 1.7344E-06 |
| $F_{18}$ | N/A | 2.6134E-04 | 3.0010E-02 | 1.7344E-06 |
| $F_{19}$ | N/A | 1.1093E-01 | 9.9179E-01 | 4.7162E-02 |
| $F_{20}$ | N/A | 1.6503E-01 | 1.5286E-01 | 1.7344E-06 |
| $F_{21}$ | N/A | 6.1564E-04 | 7.5213E-02 | 1.9209E-06 |
| $F_{22}$ | N/A | 8.2901E-01 | 3.3886E-01 | 1.7423E-04 |
| $F_{23}$ | N/A | 2.5000E-01 | 1.0000E+00 | 1.0000E+00 |
| $F_{24}$ | N/A | 2.4118E-04 | 7.8647E-02 | 2.0859E-04 |
| $F_{25}$ | N/A | 1.0000E+00 | 1.0000E+00 | 1.0000E+00 |
| $F_{26}$ | N/A | 2.5967E-05 | 2.8948E-01 | 2.8786E-06 |
| $F_{27}$ | N/A | 1.2290E-05 | 1.0000E+00 | 1.0000E+00 |
| $F_{28}$ | N/A | 2.5631E-06 | 1.0000E+00 | 1.0000E+00 |
| $F_{29}$ | N/A | 1.2453E-02 | 3.1603E-02 | 3.1123E-05 |
| $F_{30}$ | N/A | 9.7110E-05 | 1.1079E-02 | 6.7328E-01 |
| +/-/ = | ~ | 13/7/10 | 10/0/20 | 24/1/5 |
| ARV | 1.9689 | 2.4944 | 2.2961 | 3.2406 |
| Rank | 1 | 3 | 2 | 4 |

**Table 6. Parameter setting for 10 metaheuristics.**

| Method | Other parameters |
|---|---|
| BMWOA | $b = 1$; $bw = 0.001$; $Beta = 0.1$; $G = MaxFES$; |
| CBA | $C_w = 3$; $f_{max} = 2.5$ $r_1 = r_2 = 0.5+rand(0,1)$; |
| EM | $IndexLB = 0$; $LSITER = 5$; $delta = 0.1$ |
| OBSCA | $a = 1$; $r_1 = a - fes*((a)/MaxFEs)$; $r_2 = (2*pi)*rand()$; |
| IGWO | $a = 2 - FEs*(2/MaxFEs)$; $A = 2*rand(0,1)*a - a$; $C = 2*rand(0,1)$ |
| MFO | $b = 1$; $t = [-1, 1]$; $a \in [-1, -2]$ |
| ALCPSO | $\omega = 0.4$; $c_1 = c_2 = 2.0$; $\theta_0 = 60$; $T = 2$ |
| CGPSO | $w = 1$; $c_1 = 2$; $c_2 = 2$ |
| SCADE | $beta\_min = 0.2$; $beta\_max = 0.8$; $pCR = 0.8$; $a = 1$; |
| HGWO | $a \in [0, -2]$; $beta\_min = 0.2$; $beta\_max = 0.8$; $pCR = 0.8$; $a = 1$; |

**Table 7. Comparative results for SGHHO and other reported methods on the benchmark function.**

| | $F_1$ | | $F_2$ | | $F_3$ | | $F_4$ | | $F_5$ | |
|---|---|---|---|---|---|---|---|---|---|---|
| | AVG | STD | AVG | STD | AVG | STD | AVG | STD | AVG | STD |
| SGHHO | **1.086E+06** | **4.525E+05** | 1.794E+04 | 1.095E+04 | **3.466E+02** | **2.632E+01** | 4.807E+02 | 4.176E+01 | **5.200E+02** | **4.855E-02** |
| BMWOA | 1.328E+08 | 6.652E+07 | 5.999E+08 | 2.419E+08 | 5.935E+04 | 1.028E+04 | 7.335E+02 | 8.664E+01 | 5.210E+02 | 9.057E-02 |
| CBA | 4.367E+06 | 1.501E+06 | 1.274E+04 | 8.683E+03 | 3.622E+03 | 2.818E+03 | 5.052E+02 | 4.222E+01 | 5.201E+02 | 1.542E-01 |
| EM | 2.868E+07 | 7.863E+06 | 5.761E+08 | 1.110E+08 | 1.653E+04 | 4.224E+03 | 6.235E+02 | 5.650E+01 | 5.209E+02 | 4.482E-02 |
| OBSCA | 4.472E+08 | 1.244E+08 | 2.451E+10 | 5.098E+09 | 5.196E+04 | 6.644E+03 | 2.273E+03 | 7.249E+02 | 5.210E+02 | 4.437E-02 |
| IGWO | 1.735E+07 | 6.680E+06 | 2.516E+06 | 1.072E+06 | 6.642E+03 | 2.756E+03 | 5.287E+02 | 2.648E+01 | 5.205E+02 | 1.276E-01 |
| MFO | 9.473E+07 | 8.813E+07 | 1.271E+10 | 7.749E+09 | 9.362E+04 | 5.512E+04 | 1.833E+03 | 1.816E+03 | 5.203E+02 | 1.839E-01 |
| ALCPSO | 4.340E+06 | 2.918E+06 | **2.101E+03** | **2.811E+03** | 3.908E+02 | 2.213E+02 | 5.224E+02 | 4.300E+01 | 5.208E+02 | 5.500E-02 |
| CGPSO | 9.345E+06 | 2.153E+06 | 1.593E+08 | 1.619E+07 | 2.287E+03 | 4.542E+02 | **4.691E+02** | **3.582E+01** | 5.210E+02 | 4.413E-02 |
| SCADE | 4.776E+08 | 9.976E+07 | 3.008E+10 | 4.162E+09 | 5.478E+04 | 6.607E+03 | 2.345E+03 | 4.389E+02 | 5.209E+02 | 7.264E-02 |
| HGWO | 1.796E+08 | 5.143E+07 | 8.579E+09 | 1.883E+09 | 6.628E+04 | 4.914E+03 | 9.308E+02 | 6.883E+01 | 5.208E+02 | 1.093E-01 |
| | $F_6$ | | $F_7$ | | $F_8$ | | $F_9$ | | $F_{10}$ | |
| | AVG | STD | AVG | STD | AVG | STD | AVG | STD | AVG | STD |
| SGHHO | 6.206E+02 | 3.914E+00 | 7.001E+02 | 3.251E-02 | 8.420E+02 | 7.975E+00 | 1.075E+03 | 1.949E+01 | **1.501E+03** | **3.661E+02** |
| BMWOA | 6.341E+02 | 4.517E+00 | 7.057E+02 | 1.305E+00 | 9.749E+02 | 2.579E+01 | 1.132E+03 | 3.249E+01 | 5.219E+03 | 6.643E+02 |
| CBA | 6.396E+02 | 2.693E+00 | **7.000E+02** | **9.437E-03** | 1.010E+03 | 4.488E+01 | 1.183E+03 | 6.175E+01 | 5.551E+03 | 7.137E+02 |
| EM | 6.240E+02 | 2.478E+00 | 7.088E+02 | 2.053E+00 | 9.002E+02 | 1.453E+01 | 1.063E+03 | 1.789E+01 | 3.103E+03 | 5.394E+02 |
| OBSCA | 6.318E+02 | 1.320E+00 | 9.185E+02 | 3.618E+01 | 1.061E+03 | 1.842E+01 | 1.194E+03 | 1.933E+01 | 6.350E+03 | 5.035E+02 |
| IGWO | 6.199E+02 | 2.338E+00 | 7.010E+02 | 6.772E-02 | 8.845E+02 | 2.082E+01 | 1.021E+03 | 2.368E+01 | 3.348E+03 | 4.702E+02 |
| MFO | 6.244E+02 | 4.195E+00 | 8.363E+02 | 7.093E+01 | 9.478E+02 | 5.108E+01 | 1.121E+03 | 5.376E+01 | 4.309E+03 | 9.934E+02 |
| ALCPSO | **6.169E+02** | **2.935E+00** | 7.000E+02 | 2.268E-02 | **8.242E+02** | **1.068E+01** | 9.993E+02 | 2.213E+01 | 1.542E+03 | 3.599E+02 |
| CGPSO | 6.248E+02 | 2.871E+00 | 7.024E+02 | 1.693E-01 | 9.879E+02 | 2.013E+01 | 1.123E+03 | 2.248E+01 | 5.565E+03 | 6.705E+02 |
| SCADE | 6.341E+02 | 3.013E+00 | 9.135E+02 | 3.852E+01 | 1.067E+03 | 1.678E+01 | 1.207E+03 | 1.723E+01 | 7.397E+03 | 3.385E+02 |
| HGWO | 6.264E+02 | 1.716E+00 | 7.464E+02 | 1.277E+01 | 1.008E+03 | 1.311E+01 | 1.140E+03 | 1.135E+01 | 5.559E+03 | 3.283E+02 |
| | $F_{11}$ | | $F_{12}$ | | $F_{13}$ | | $F_{14}$ | | $F_{15}$ | |
| | AVG | STD | AVG | STD | AVG | STD | AVG | STD | AVG | STD |
| SGHHO | 4.601E+03 | 5.183E+02 | **1.200E+03** | **1.434E-01** | **1.300E+03** | **9.774E-02** | **1.400E+03** | **4.411E-02** | **1.509E+03** | **2.665E+00** |
| BMWOA | 7.192E+03 | 5.139E+02 | 1.202E+03 | 4.884E-01 | 1.301E+03 | 1.220E-01 | 1.400E+03 | 1.845E-01 | 1.625E+03 | 7.009E+01 |
| CBA | 5.653E+03 | 7.579E+02 | 1.201E+03 | 5.669E-01 | 1.300E+03 | 1.295E-01 | 1.400E+03 | 1.858E-01 | 1.565E+03 | 1.679E+01 |
| EM | 4.839E+03 | 4.537E+02 | 1.201E+03 | 4.240E-01 | 1.300E+03 | 5.764E-02 | 1.400E+03 | 2.608E-02 | 1.531E+03 | 3.685E+00 |
| OBSCA | 7.284E+03 | 3.851E+02 | 1.202E+03 | 3.833E-01 | 1.304E+03 | 3.701E-01 | 1.472E+03 | 1.598E+01 | 1.481E+04 | 6.757E+03 |
| IGWO | 4.401E+03 | 7.100E+02 | 1.201E+03 | 4.191E-01 | 1.301E+03 | 1.118E-01 | 1.400E+03 | 3.082E-01 | 1.517E+03 | 5.274E+00 |
| MFO | 5.250E+03 | 7.927E+02 | 1.200E+03 | 2.517E-01 | 1.302E+03 | 1.398E+00 | 1.435E+03 | 2.782E+01 | 8.370E+04 | 1.388E+05 |
| ALCPSO | **4.137E+03** | **5.398E+02** | 1.201E+03 | 3.961E-01 | 1.301E+03 | 9.528E-02 | 1.401E+03 | 3.066E-01 | 1.511E+03 | 3.216E+00 |
| CGPSO | 5.803E+03 | 5.953E+02 | 1.202E+03 | 3.131E-01 | 1.300E+03 | 8.783E-02 | 1.400E+03 | 1.130E-01 | 1.518E+03 | 1.330E+00 |
| SCADE | 8.162E+03 | 2.648E+02 | 1.203E+03 | 3.130E-01 | 1.304E+03 | 2.754E-01 | 1.488E+03 | 1.520E+01 | 1.759E+04 | 6.057E+03 |
| HGWO | 6.535E+03 | 4.576E+02 | 1.201E+03 | 2.522E-01 | 1.302E+03 | 4.605E-01 | 1.422E+03 | 3.569E+00 | 1.934E+03 | 3.043E+02 |
| | $F_{16}$ | | $F_{17}$ | | $F_{18}$ | | $F_{19}$ | | $F_{20}$ | |
| | AVG | STD | AVG | STD | AVG | STD | AVG | STD | AVG | STD |
| SGHHO | **1.612E+03** | **5.177E-01** | **1.463E+05** | **9.728E+04** | **3.516E+03** | **1.919E+03** | **1.916E+03** | **3.787E+00** | **2.290E+03** | **5.894E+01** |
| BMWOA | 1.613E+03 | 2.869E-01 | 6.923E+06 | 4.111E+06 | 5.517E+05 | 1.254E+06 | 1.952E+03 | 3.466E+01 | 4.331E+04 | 3.720E+04 |
| CBA | 1.613E+03 | 3.189E-01 | 2.449E+05 | 1.928E+05 | 8.575E+03 | 8.575E+03 | 1.941E+03 | 4.039E+01 | 3.470E+03 | 2.426E+03 |
| EM | 1.612E+03 | 4.422E-01 | 7.659E+05 | 1.916E+05 | 7.856E+07 | 6.276E+07 | 1.943E+03 | 3.484E+01 | 2.726E+03 | 1.957E+02 |
| OBSCA | 1.613E+03 | 1.832E-01 | 1.054E+07 | 4.426E+06 | 1.569E+08 | 1.014E+08 | 2.007E+03 | 2.367E+01 | 3.371E+04 | 1.525E+04 |
| IGWO | 1.612E+03 | 5.580E-01 | 1.043E+06 | 5.466E+05 | 2.039E+04 | 2.097E+04 | 1.924E+03 | 2.388E+01 | 3.286E+03 | 1.047E+03 |
| MFO | 1.613E+03 | 7.325E-01 | 3.342E+06 | 5.364E+06 | 4.243E+07 | 1.490E+08 | 1.976E+03 | 6.124E+01 | 6.043E+04 | 2.962E+04 |

*(Continued)*

**Table 7.** (Continued)

|  | $F_1$ |  | $F_2$ |  | $F_3$ |  | $F_4$ |  | $F_5$ |  |
|---|---|---|---|---|---|---|---|---|---|---|
|  | AVG | STD | AVG | STD | AVG | STD | AVG | STD | AVG | STD |
| ALCPSO | 1.612E+03 | 4.627E-01 | 5.542E+05 | 5.301E+05 | 8.337E+03 | 6.019E+03 | 1.918E+03 | 2.354E+01 | 3.111E+03 | 6.863E+02 |
| CGPSO | 1.612E+03 | 4.590E-01 | 3.628E+05 | 1.930E+05 | 2.436E+06 | 6.933E+05 | 1.917E+03 | 2.717E+00 | 2.455E+03 | 1.055E+02 |
| SCADE | 1.613E+03 | 2.184E-01 | 1.534E+07 | 8.361E+06 | 1.753E+08 | 1.209E+08 | 2.013E+03 | 1.193E+01 | 2.640E+04 | 1.239E+04 |
| HGWO | 1.613E+03 | 2.712E-01 | 6.067E+06 | 2.778E+06 | 1.196E+08 | 3.500E+07 | 1.988E+03 | 1.025E+01 | 6.563E+04 | 3.160E+04 |

|  | $F_{21}$ |  | $F_{22}$ |  | $F_{23}$ |  | $F_{24}$ |  | $F_{25}$ |  |
|---|---|---|---|---|---|---|---|---|---|---|
|  | AVG | STD | AVG | STD | AVG | STD | AVG | STD | AVG | STD |
| SGHHO | **6.176E+04** | **3.141E+04** | 2.738E+03 | 1.719E+02 | **2.500E+03** | **0.000E+00** | **2.600E+03** | 4.603E-03 | **2.700E+03** | **0.000E+00** |
| BMWOA | 1.655E+06 | 1.812E+06 | 2.957E+03 | 2.393E+02 | 2.501E+03 | 6.967E-01 | 2.600E+03 | 1.442E-01 | 2.700E+03 | 9.742E-03 |
| CBA | 1.050E+05 | 7.051E+04 | 3.464E+03 | 3.632E+02 | 2.616E+03 | 2.308E-01 | 2.684E+03 | 4.089E+01 | 2.732E+03 | 1.379E+01 |
| EM | 1.685E+05 | 9.812E+04 | 2.811E+03 | 1.953E+02 | 2.572E+03 | 4.072E+01 | 2.604E+03 | 4.159E-01 | 2.701E+03 | 9.181E-02 |
| OBSCA | 2.434E+06 | 1.594E+06 | 3.153E+03 | 1.369E+02 | 2.680E+03 | 1.701E+01 | 2.600E+03 | 4.710E-04 | 2.700E+03 | 1.642E-03 |
| IGWO | 3.287E+05 | 2.600E+05 | **2.537E+03** | **1.604E+02** | 2.621E+03 | 2.823E+00 | 2.600E+03 | 5.320E-03 | 2.709E+03 | 1.685E+00 |
| MFO | 1.285E+06 | 2.537E+06 | 3.037E+03 | 2.217E+02 | 2.659E+03 | 3.606E+01 | 2.683E+03 | 2.886E+01 | 2.715E+03 | 7.177E+00 |
| ALCPSO | 8.151E+04 | 8.307E+04 | 2.682E+03 | 1.795E+02 | 2.615E+03 | 4.024E-02 | 2.638E+03 | 7.279E+00 | 2.711E+03 | 3.859E+00 |
| CGPSO | 1.369E+05 | 9.974E+04 | 2.919E+03 | 2.370E+02 | 2.500E+03 | 2.528E-03 | 2.600E+03 | 9.937E-03 | 2.700E+03 | 2.201E-05 |
| SCADE | 2.792E+06 | 1.574E+06 | 3.122E+03 | 1.488E+02 | 2.500E+03 | 0.000E+00 | 2.600E+03 | 1.865E-06 | 2.700E+03 | 0.000E+00 |
| HGWO | 2.214E+06 | 1.608E+06 | 3.007E+03 | 1.170E+02 | 2.522E+03 | 5.759E+01 | 2.600E+03 | 0.000E+00 | 2.700E+03 | 0.000E+00 |

|  | $F_{26}$ |  | $F_{27}$ |  | $F_{28}$ |  | $F_{29}$ |  | $F_{30}$ |  |
|---|---|---|---|---|---|---|---|---|---|---|
|  | AVG | STD | AVG | STD | AVG | STD | AVG | STD | AVG | STD |
| SGHHO | 2.704E+03 | 1.819E+01 | **2.900E+03** | **0.000E+00** | **3.000E+03** | **0.000E+00** | **3.109E+03** | **4.734E+00** | 3.500E+03 | 3.934E+02 |
| BMWOA | **2.701E+03** | **1.442E-01** | 2.900E+03 | 1.096E-01 | 3.000E+03 | 2.858E-01 | 9.269E+05 | 2.066E+06 | 3.734E+04 | 3.379E+04 |
| CBA | 2.711E+03 | 5.526E+01 | 4.043E+03 | 3.863E+02 | 5.597E+03 | 8.197E+02 | 4.368E+07 | 4.752E+07 | 2.272E+04 | 2.818E+04 |
| EM | 2.788E+03 | 3.144E+01 | 3.641E+03 | 2.867E+02 | 5.798E+03 | 1.336E+03 | 1.024E+07 | 1.936E+07 | 1.089E+04 | 3.102E+03 |
| OBSCA | 2.704E+03 | 5.668E-01 | 3.249E+03 | 4.240E+01 | 5.512E+03 | 3.591E+02 | 2.242E+07 | 1.211E+07 | 3.806E+05 | 1.293E+05 |
| IGWO | 2.701E+03 | 1.578E-01 | 3.109E+03 | 4.281E+00 | 3.797E+03 | 9.108E+01 | 2.356E+06 | 4.836E+06 | 2.823E+04 | 1.190E+04 |
| MFO | 2.703E+03 | 1.141E+00 | 3.607E+03 | 2.188E+02 | 3.914E+03 | 2.249E+02 | 2.048E+06 | 3.486E+06 | 4.859E+04 | 4.239E+04 |
| ALCPSO | 2.747E+03 | 5.066E+01 | 3.399E+03 | 2.389E+02 | 4.546E+03 | 4.293E+02 | 3.718E+06 | 7.262E+06 | 1.410E+04 | 1.036E+04 |
| CGPSO | 2.797E+03 | 1.820E+01 | 3.014E+03 | 2.960E+02 | 3.000E+03 | 9.194E-03 | 5.970E+03 | 3.022E+03 | 1.113E+04 | 8.882E+03 |
| SCADE | 2.707E+03 | 1.758E+01 | 3.242E+03 | 1.982E+02 | 4.906E+03 | 1.018E+03 | 1.637E+07 | 7.805E+06 | 4.419E+05 | 1.544E+05 |
| HGWO | 2.743E+03 | 4.926E+01 | 3.552E+03 | 2.675E+02 | 4.214E+03 | 2.270E+02 | 3.768E+06 | 3.657E+06 | **3.200E+03** | **1.731E-04** |

optimal, there is no significant difference compared to the optimal solution for each function and the average AVG ranking is better. On the hybrid functions, SGHHO can search for optimal solutions for most functions compared to other advanced algorithms. A comparison to this can be clearly seen in the F10-F20 functions. Also, using the Friedman test, this study compares the combined strength of SGHHO and these competing algorithms.

As the data at the end of Table 8 indicates, SGHHO achieves the first overall average ranking in processing the CEC 2014 function set, followed by algorithms including ALCPSO, IGWO and CGPSO. Besides, according to the Wilcoxon test, the p-value of SGHHO is mostly less than 0.05 compared to other methods, which demonstrates the significant difference between the SGHHO algorithm compared to other competitors. It validates that the SGHHO algorithm, which incorporates the Gaussian Bare-Bones strategy and the simulated annealing strategy, is a more promising algorithm for different types of optimization problems, and that SGHHO significantly improves HHO's optimization power.

Fig 3 illustrates the convergence curves of SGHHO in comparison to other state-of-the-art algorithms on the CEC 2014 test suite. The convergence curves provide a more intuitive

**Table 8. P-values for SGHHO and other reported algorithms by Wilcoxon test on the benchmark function.**

| Function | SGHHO | BMWOA | CBA | EM | OBSCA | IGWO | MFO | ALCPSO | CGPSO | SCADE | HGWO |
|---|---|---|---|---|---|---|---|---|---|---|---|
| $F_1$ | N/A | 1.7E-06 | 1.7E-06 | 1.7E-06 | 1.7E-06 | 1.7E-06 | 1.7E-06 | 1.8E-05 | 1.7E-06 | 1.7E-06 | 1.7E-06 |
| $F_2$ | N/A | 1.7E-06 | **7.2E-02** | 1.7E-06 | 1.7E-06 | 1.7E-06 | 1.7E-06 | 2.4E-06 | 1.7E-06 | 1.7E-06 | 1.7E-06 |
| $F_3$ | N/A | 1.7E-06 | 1.7E-06 | 1.7E-06 | 1.7E-06 | 1.7E-06 | 1.7E-06 | **1.6E-01** | 1.7E-06 | 1.7E-06 | 1.7E-06 |
| $F_4$ | N/A | 1.7E-06 | 2.3E-02 | 1.7E-06 | 1.7E-06 | 5.3E-05 | 1.7E-06 | 8.9E-04 | **3.3E-01** | 1.7E-06 | 1.7E-06 |
| $F_5$ | N/A | 1.7E-06 | **7.036E-01** | 1.7E-06 | 1.7E-06 | 1.7E-06 | 3.2E-06 | 1.7E-06 | 1.7E-06 | 1.7E-06 | 1.7E-06 |
| $F_6$ | N/A | 1.7E-06 | 1.7E-06 | 1.2E-03 | 1.7E-06 | **3.8E-01** | 6.0E-03 | 1.2E-03 | 2.6E-05 | 1.7E-06 | 1.7E-06 |
| $F_7$ | N/A | 1.7E-06 | 1.7E-06 | 1.7E-06 | 1.7E-06 | 1.7E-06 | 1.7E-06 | 1.1E-05 | 1.7E-06 | 1.7E-06 | 1.7E-06 |
| $F_8$ | N/A | 1.7E-06 | 1.7E-06 | 1.7E-06 | 1.7E-06 | 1.9E-06 | 1.7E-06 | 1.4E-05 | 1.7E-06 | 1.7E-06 | 1.7E-06 |
| $F_9$ | N/A | 2.9E-06 | 1.9E-06 | 2.5E-02 | 1.7E-06 | 3.2E-06 | 2.1E-04 | 1.7E-06 | 4.3E-06 | 1.7E-06 | 1.7E-06 |
| $F_{10}$ | N/A | 1.7E-06 | 1.7E-06 | 1.9E-06 | 1.7E-06 | 1.7E-06 | 1.7E-06 | **9.1E-01** | 1.7E-06 | 1.7E-06 | 1.7E-06 |
| $F_{11}$ | N/A | 1.7E-06 | 2.2E-05 | 4.5E-02 | 1.7E-06 | **1.4E-01** | 2.3E-03 | 3.6E-03 | 2.4E-06 | 1.7E-06 | 1.7E-06 |
| $F_{12}$ | N/A | 1.7E-06 | 2.4E-06 | 2.4E-06 | 1.7E-06 | 2.9E-03 | **5.6E-01** | 2.6E-06 | 1.7E-06 | 1.7E-06 | 1.7E-06 |
| $F_{13}$ | N/A | 1.7E-06 | 1.5E-05 | 3.3E-04 | 1.7E-06 | 1.9E-06 | 1.7E-06 | 2.6E-06 | 1.1E-05 | 1.7E-06 | 1.7E-06 |
| $F_{14}$ | N/A | 1.7E-06 | 2.4E-06 | **9.1E-01** | 1.7E-06 | 1.4E-05 | 1.7E-06 | 1.734E-06 | 4.5E-04 | 1.7E-06 | 1.7E-06 |
| $F_{15}$ | N/A | 1.7E-06 | 1.7E-06 | 1.7E-06 | 1.7E-06 | 2.6E-06 | 1.9E-06 | **8.6E-02** | 1.7E-06 | 1.7E-06 | 1.7E-06 |
| $F_{16}$ | N/A | 2.3E-06 | 1.7E-06 | 2.6E-03 | 1.7E-06 | **2.2E-01** | 9.3E-06 | **4.0E-01** | **1.6E-01** | 1.7E-06 | 1.7E-06 |
| $F_{17}$ | N/A | 1.7E-06 | 1.2E-02 | 1.7E-06 | 1.7E-06 | 1.734E-06 | 2.8E-05 | 4.2E-04 | 3.1E-05 | 1.7E-06 | 1.7E-06 |
| $F_{18}$ | N/A | 1.7E-06 | 8.3E-04 | 1.7E-06 | 1.7E-06 | 8.5E-06 | 2.3E-06 | 3.8E-04 | 1.7E-06 | 1.7E-06 | 1.7E-06 |
| $F_{19}$ | N/A | 1.9E-06 | 6.6E-04 | 2.6E-02 | 1.7E-06 | **1.5E-01** | 2.8E-05 | 2.2E-02 | **2.8E-01** | 1.7E-06 | 1.7E-06 |
| $F_{20}$ | N/A | 1.7E-06 | 4.7E-06 | 1.7E-06 | 1.7E-06 | 1.7E-06 | 1.7E-06 | 2.6E-06 | 2.3E-06 | 1.7E-06 | 1.7E-06 |
| $F_{21}$ | N/A | 1.7E-06 | 1.6E-03 | 9.3E-06 | 1.7E-06 | 1.7E-06 | 1.6E-05 | **4.9E-01** | 3.7E-05 | 1.7E-06 | 1.7E-06 |
| $F_{22}$ | N/A | 4.9E-04 | 2.3E-06 | **1.3E-01** | 1.7E-06 | 1.7E-04 | 5.8E-06 | **2.1E-01** | 3.6E-03 | 3.2E-06 | 4.7E-06 |
| $F_{23}$ | N/A | 1.7E-06 | 1.7E-06 | 1.7E-06 | 1.7E-06 | 1.7E-06 | 1.7E-06 | 1.7E-06 | 1.7E-06 | **1.0E+00** | 6.3E-02 |
| $F_{24}$ | N/A | 1.7E-06 | 1.7E-06 | 1.7E-06 | **8.9E-02** | 3.3E-04 | 1.7E-06 | 1.7E-06 | 1.7E-06 | 1.1E-05 | 3.8–06 |
| $F_{25}$ | N/A | 1.7E-06 | 1.7E-06 | 1.7E-06 | **6.3E-02** | 1.7E-06 | 1.7E-06 | 1.7E-06 | 1.7E-06 | **1.0E+00** | **1.0E+00** |
| $F_{26}$ | N/A | 7.5E-05 | 2.8E-02 | 1.7E-06 | 3.112E-05 | 3.4E-05 | 3.112E-05 | 2.163E-05 | 2.1E-06 | 2.8E-05 | 1.1E-05 |
| $F_{27}$ | N/A | 1.7E-06 | 1.7E-06 | 1.7E-06 | 1.7E-06 | 1.7E-06 | 1.7E-06 | 1.7E-06 | 1.7E-06 | 1.8E-05 | 2.6E-06 |
| $F_{28}$ | N/A | 1.7E-06 | 1.7E-06 | 1.7E-06 | 1.7E-06 | 1.7E-06 | 1.7E-06 | 1.7E-06 | 1.7E-06 | 1.8E-05 | 1.7E-06 |
| $F_{29}$ | N/A | 1.7E-06 | 1.7E-06 | 1.7E-06 | 1.7E-06 | 1.7E-06 | 1.7E-06 | 1.7E-06 | 1.7E-06 | 1.7E-06 | 1.6E-04 |
| $F_{30}$ | N/A | 1.7E-06 | 1.7E-06 | 1.7E-06 | 1.7E-06 | 1.7E-06 | 1.7E-06 | 1.7E-06 | 2.6E-04 | 1.7E-06 | 1.7E-06 |
| +/-/ = | ~ | 29/1/0 | 27/1/2 | 27/1/2 | 28/0/2 | 23/3/4 | 28/1/1 | 19/5/6 | 27/0/3 | 27/1/2 | 26/2/2 |
| ARV | **2.1461** | 6.8489 | 6.0111 | 5.7344 | 9.0017 | 4.5122 | 7.1156 | 3.9567 | 4.9189 | 8.7606 | 6.9939 |
| Rank | **1** | 7 | 6 | 5 | 11 | 3 | 9 | 2 | 4 | 10 | 8 |

picture of the difference in convergence quality and optimization capability between SGHHO and the competitors. The figure clearly demonstrates that SGHHO exhibits faster convergence rates for the majority of functions. For example, F1, F10, F16, F29 and F30. This is because SGHHO utilizes the Gaussian Bare Bones strategy for individual variation and can expand the global search early in the evolutionary process. Furthermore, an observation can be made from the results of F3, F5, F18, and F21, which indicate that even though SGHHO may not exhibit the fastest early convergence, it manages to sustain its progress towards the optimal value after ALPSO, CBA, and CGPSO have reached convergence. This signifies the ability of SGHHO to effectively prevent being trapped in local optima. Also, SGHHO converges rapidly in the later evolutionary stage through SA mechanism, and eventually achieves higher precision and better solutions in the search process when compared with other comparative algorithms. This makes algorithms such as OBSCA, ALCPSO, BMWOA and SCADE fall into premature convergence, while SGHHO can maintain the convergence trend until the global

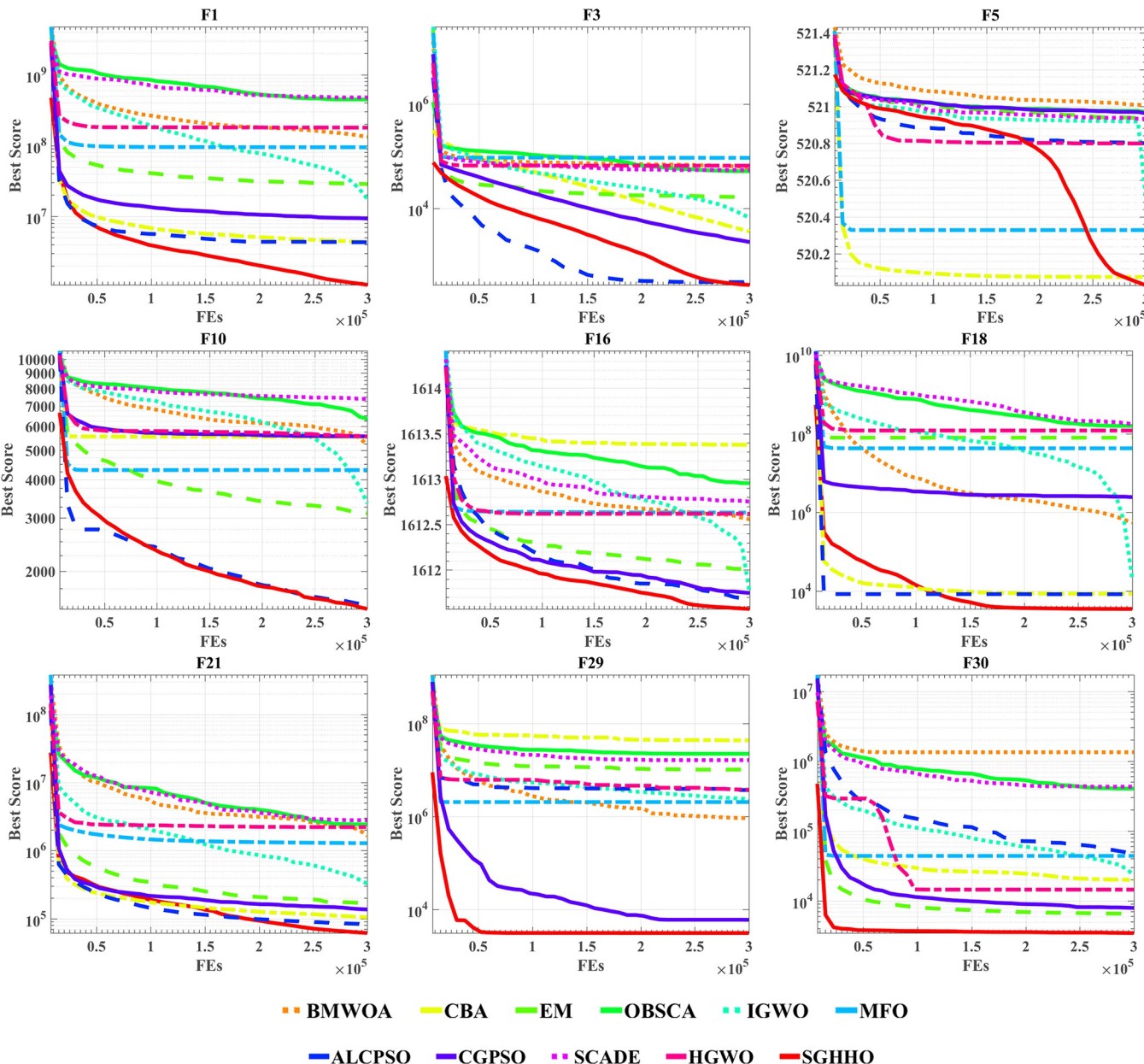

**Fig 3. Convergence curves for SGHHO and advanced algorithms on 9 selected benchmark functions.**

optimum. SGHHO demonstrates a remarkable capability to effectively maintain a balance between exploration and exploitation throughout the search process, thereby rejuvenating the evolution of the population. This ability proves instrumental in mitigating the issue of premature convergence that is commonly encountered in HHO.

A comparison of the mean, standard deviation values, Wilcoxon test and convergence figures leads to the conclusion that the proposed SGHHO significantly outperforms other competing algorithms. It can consistently maintain a leading position in function optimization problems, and the advantage of SGHHO becomes more obvious as the dimensionality of the problem increases.

Hence, in subsequent investigations, this approach can be extended to a broader range of scenarios, such as optimization of machine learning models [120], computer-aided medical diagnosis [121,122], pathology image segmentation [123–125], image denoising [126,127], fine-grained alignment [128], cancer diagnosis [129–131], medical signals [132,133], and structured sparsity optimization [134].

# 7. Predicting the future of the sharing economy

## 7.1. Methods & data collection

The sharing economy dataset is collected for this study and deals with data mainly from participants in the general public who have participated in the sharing economy. From these actual participants, 671 participants were selected for the study. The questionnaire was developed with reference to numerous scholars' questionnaires on the sharing economy and predicted to have good reliability and validity. The questionnaire analyses 43 aspects of the respondents by examining their gender, education level, political status, age, profession, location, income level, consumption level, various forms of sharing economy and their perception of sharing economy (see Table 9). This study examines the basic situation of the public's participation in sharing economy, explores the importance of these attributes and their inherent linkages, and builds a predictive model of future trends in the sharing economy on this basis.

Since the above-mentioned questionnaires did not involve ethical issues, the review committee/ethics committee of Wenzhou University granted an exemption from ethical review. All participants in the questionnaires signed a consent form.

## 7.2. Condition configuration

The experiments involved in this work were completed on a host computer with 16GB of RAM, 3.6GHz main frequency, Windows 10 and coded on MATLAB R2016b software. In addition, the experimental parameters are extremely important to the simulation and subtle changes in the parameters can affect the final experimental outcome. To ensure the fairness of the study, the proposed models were quantitatively evaluated using statistical techniques including the calculation of average values (AVG) and standard deviations (STD). The AVG represents the average predictive performance of each model, while the STD indicates the degree of variation across 10 independent runs. To further assess the effectiveness of the proposed models, four commonly used classification metrics were employed, namely Accuracy (ACC), Sensitivity, Specificity, and Matthew's correlation coefficient (MCC).

## 7.3. Experimental results and analysis of SGHHO-KNN

To investigate the key factors influencing the future development of sharing economy, this experiment uses a variety of machine learning models to make predictions from a real dataset collected, including BSGHHO-KNN, BSHHO-KNN, BGHHO-KNN, BP, RF, ELM and other classification models. Meanwhile, the classification results of each model will be statistically analyzed using four indicators, namely ACC, Sensitivity, Specificity and MCC. Furthermore, the comprehensive experimental findings are disclosed in Table 10, providing detailed insights into the results obtained. And it can be intuitively found that the BSGHHO-KNN model obtained 99.70%—the highest value of accuracy in predicting the future development trend of sharing economy, while the other five models were 98.66%, 93.45%, 51.56%, 99.11% and 65.73% respectively. While the BSHHO-KNN and RF models demonstrated satisfactory accuracy, it is worth noting that the BSGHHO-KNN model exhibited superior performance in terms of the Specificity and MCC metrics, surpassing both of them and attaining the highest

**Table 9. Descriptions of each attribute.**

| ID | Attributes | Description |
|---|---|---|
| F1 | Gender | Male and female participants are represented by 1 and 2 respectively. |
| F2 | Education level | The categories are classified into various levels of education, including doctoral, master's, bachelor's, college, and high school or below. These levels are denoted by numerical values 1, 2, 3, 4, and 5, respectively. |
| F3 | Political status | The classification is categorized into four groups: members of the Communist Party, reserve party members, members of the Communist Youth League, and the general population. These categories are symbolized by the numerical values 1, 2, 3, and 4, respectively. |
| F4 | Age | It is divided into under 15 years of age, 15–30 years of age, 31–35 years of age, 46–60 years of age, and 61 years of age and above denoted by 1, 2, 3, 4 and 5 respectively. |
| F5 | Professional category | It is divided into law, engineering, management, education, economics, science, literature, medicine and art, indicated by 1, 2, 3, 4, 5, 6, 7, 8 and 9 respectively. |
| F6 | Region | It is divided into North China, East China, Central China, South China, Northeast China, represented by 1, 2, 3, 4, 5, 6 and 7 respectively. |
| F7 | Form of sharing economy in your region | Multiple choice types of car sharing, bicycle sharing, electric bicycle sharing and umbrella sharing. |
| F8 | Monthly consumption level | Classified as below RMB1,000, RMB1,000–3,000, RMB3,001–5,000, RMB5,001–8,000, and above RMB8,001, represented by 1, 2, 3, 4 & 5 respectively. |
| F9 | Annual household income | Divided into less than $50,000, $50,000-$100,000...$400,000-$500,000, and over $500,000 represented by 1, 2, 3, 4, 5, 6, and 7 respectively. |
| F10 | Occupation | The categories are students, teachers, doctors, civil servants, company employees, and others, represented by 1, 2, 3, 4, 5, 6, 7 and 8 respectively. |
| F11 | Awareness of the sharing economy | There are five categories: not at all, basically, extremely, researched and researched in depth, represented by 1, 2, 3, 4 and 5 respectively. |
| F12 | Awareness of the sharing economy | The response options are 1, 2, 3, 4 and 5 respectively. |
| F13 | How did you first learn about the sharing economy? | Media coverage, shared equipment, books and articles, people around, and other are represented by 1, 2, 3, 4, and 5 respectively. |
| F14 | Whether or not you have used sharing economy services | No, yes, indicated by 0, 1 respectively. |
| F15 | Frequency of use of shared products | Not at all, Occasional use, Regular use, indicated by 1, 2, 3 respectively. |
| F16 | How much you pay for sharing economy products | Cheap, fair, barely acceptable, expensive, indicated by 1, 2, 3, 4 respectively. |
| F17 | Willingness to learn about the sharing economy | No, yes, indicated by 0, 1 respectively. |
| F18 | Whether they are excluded from using shared products and services | No, yes, indicated by 0, 1 respectively. |
| F19 | Willingness to participate in the sharing economy as a provider | No, yes, indicated by 0, 1 respectively. |
| F20 | Perceived most important features of the sharing economy | Response options are indicated by 1, 2, 3, 4 and 5 respectively. |
| F21 | Perception of which consumer group the sharing product would be most helpful to | The categories are students, teachers, doctors, civil servants, company employees, management personnel, freelancers and others, represented by 1, 2, 3, 4, 5, 6, 7 and 8 respectively. |

(*Continued*)

**Table 9.** (Continued）

| ID | Attributes | Description |
|----|-----------|-------------|
| F22 | Products that have been used in the sharing sector | Car-sharing, bicycle-sharing, electric bicycle-sharing, umbrella-sharing and other multiple-choice types. |
| F23 | Factors that concern you most when using shared product services | Quality of service, price of service, ease of access to service, personal privacy issues, personal safety issues, other, represented by 1, 2, 3, 4, 5, 6 respectively. |
| F24 | Impact on life and work | No impact, low impact, average impact, high impact, high impact, indicated by 1, 2, 3, 4 respectively. |
| F25 | Whether the sharing economy has brought convenience | No, yes, indicated by 0, 1 respectively. |
| F26 | Key aspects of convenience brought about by the sharing economy | Price, time, efficiency, convenience, experience, indicated by 1, 2, 3, 4, 5 respectively. |
| F27 | Whether or not the sharing economy is a nuisance | No, yes, indicated by 0, 1 respectively. |
| F28 | Key aspects of distress caused by the sharing economy | Answer options are indicated by 1, 2, 3, 4, 5, 6, 7 respectively. |
| F29 | Causes of distress in the sharing economy | Answer options are indicated by 1, 2, 3, 4, 5, 6, 7 respectively. |
| F30 | Attitudes to the sharing economy | Answer options are denoted by 1, 2, 3, 4, 5 respectively |
| F31 | The main negative aspects of the development of the sharing economy in the region | Answer options are indicated by 1, 2, 3, 4, 5 |
| F32 | Positive factors for the development of the sharing economy in the region | Answer options are indicated by 1, 2, 3, 4 and 5 respectively |
| F33 | Perception of the future of "sharing economy" | Very unfavorable, rather unfavorable, average, rather favorable, very favorable, indicated by 1, 2, 3, 4, 5 respectively |
| F34 | Factors contributing to consumer demand for shared products | Risk aversion, supply shocks, income instability, demand contraction, expectations weakening, other indicated by 1, 2, 3, 4, 5, 6 respectively |
| F35 | Main concerns about the sharing economy | The available choices for answers are as follows by 1, 2, 3, 4, 5, 6, 7 respectively |
| F36 | Confidence in the sharing economy | The available choices for answers are as follows by 1, 2, 3, 4, 5, 6 respectively. |
| F37 | Perceived advantages of the sharing economy | The available choices for answers are as follows by 1, 2, 3, 4, 5, 6, 7 and 8 respectively. |
| F38 | Positive impacts of the sharing economy on society | The available choices for answers are as follows by 1, 2, 3, 4, 5, 6, 7 respectively |
| F39 | Negative effects of the sharing economy on society | The available choices for answers are as follows by 1, 2, 3, 4 respectively |
| F40 | Overall social effects of the sharing economy | The disadvantages outweigh the advantages and the advantages outweigh the disadvantages are represented by 0 and 1 respectively. |
| F41 | Ways in which people can contribute to the sharing economy | Active promotion of a sharing atmosphere, acceptance of the sharing economy with an open mind, active participation, others indicated by 1, 2, 3, 4 respectively |
| F42 | Views on the future of sharing economy | 0, 1 for only a temporary wave, 1 for a wider range of developments |
| F43 | Any suggestions for the development of the sharing economy | Answer options are 1, 2, 3, 4, 5 and 6 respectively |

classification outcomes. Based on the above four evaluation metrics it can be fully demonstrated that the BSGHHO-KNN model is feasible to be applied to the problem of predicting future trends in the sharing economy.

Table 11 shows the standard deviation values derived from the BSGHHO-KNN model and the other five models after 10 independent experiments. The optimal values in the table are

**Table 10. Average results of BSGHHO-KNN and other models on four indicators.**

| Models | ACC | Sensitivity | Specificity | MCC |
|---|---|---|---|---|
| | Avg | | | |
| **BSGHHO-KNN** | **99.70%** | **100.00%** | **99.38%** | **99.42%** |
| BSHHO-KNN | 98.66% | 100.00% | 97.19% | 97.52% |
| BGHHO-KNN | 93.45% | 98.01% | 88.44% | 87.31% |
| BP | 51.56% | 16.29% | 90.00% | 10.69% |
| RF | 99.11% | 98.86% | 99.38% | 98.24% |
| ELM | 65.73% | 71.50% | 59.38% | 31.50% |

bolded for more visual analysis of the experimental data. After analyzing the results, it can be deduced that the BSGHHO-KNN model demonstrates exceptional stability in terms of ACC, sensitivity, and MCC, surpassing other models in these aspects. In terms of specificity indicators, RF model gains the best value.

To better compare the performance gap between the models, Fig 4 presents the histograms of AVG and STD values for the five models mentioned above. The examination of Fig 4 reveals that the BSGHHO-KNN model exhibits superior performance across all four indicators, namely ACC, Sensitivity, Specificity, and MCC, resulting in the most effective classification outcome. Following closely behind are the RF model and the BSHHO-KNN model. The figure shows that BSHHO-KNN and BGHHO-KNN both have been effective compared to the other classifiers. The experiments show that the HHO algorithm has excellent adaptability in combination with the KNN classifier. In addition, the BP model performed the worst on the sharing economy dataset. The ACC value was only 51.56% and the Sensitivity was 16.29%. This suggests that the BP model needs to be tuned with appropriate parameters for the specific problem. In addition, the BSHHO-KNN model and the BGHHO-KNN model do not perform the best on the dataset, which means that the BSGHHO algorithm can enable the KNN classifier to maximize its classification performance and thus achieve better classification results. In summary, the BSGHHO-KNN model outperforms other similar methods and can be used to investigate the key factors affecting future trends in the sharing economy.

In this experimental study, the proposed model successfully accomplishes the task of selecting the optimal subset of features during the entire process. Fig 5 counts the selected times of each feature in each experiment in the form of a line graph. From the figure: "form of sharing economy in the region" (F7), "attitude towards the sharing economy" (F30), "products in the sharing domain that have been used" (F22), "Factors of greatest concern when using shared product services" (F23), "Main aspects of distress caused by the sharing economy" (F28), "Negative effects of the sharing economy on society " (F39), "Main aspects of concerns about the sharing economy" (F35) and "Any suggestions for the development of the sharing economy"

**Table 11. Standard deviation values for BSGHHO-KNN and the other five models on the four indicators.**

| Models | ACC | Sensitivity | Specificity | MCC |
|---|---|---|---|---|
| | Std | | | |
| **BSGHHO-KNN** | **9.44E-03** | **0.00E+00** | 1.98E-02 | **1.84E-02** |
| BSHHO-KNN | 3.75E-02 | 0.00E+00 | 7.86E-02 | 6.87E-02 |
| BGHHO-KNN | 3.30E-02 | 1.92E-02 | 6.76E-02 | 6.30E-02 |
| BP | 6.08E-02 | 3.16E-01 | 3.16E-01 | 1.62E-01 |
| RF | 1.26E-02 | 2.00E-02 | **1.32E-02** | 2.48E-02 |
| ELM | 7.55E-02 | 1.15E-01 | 1.02E-01 | 1.52E-01 |

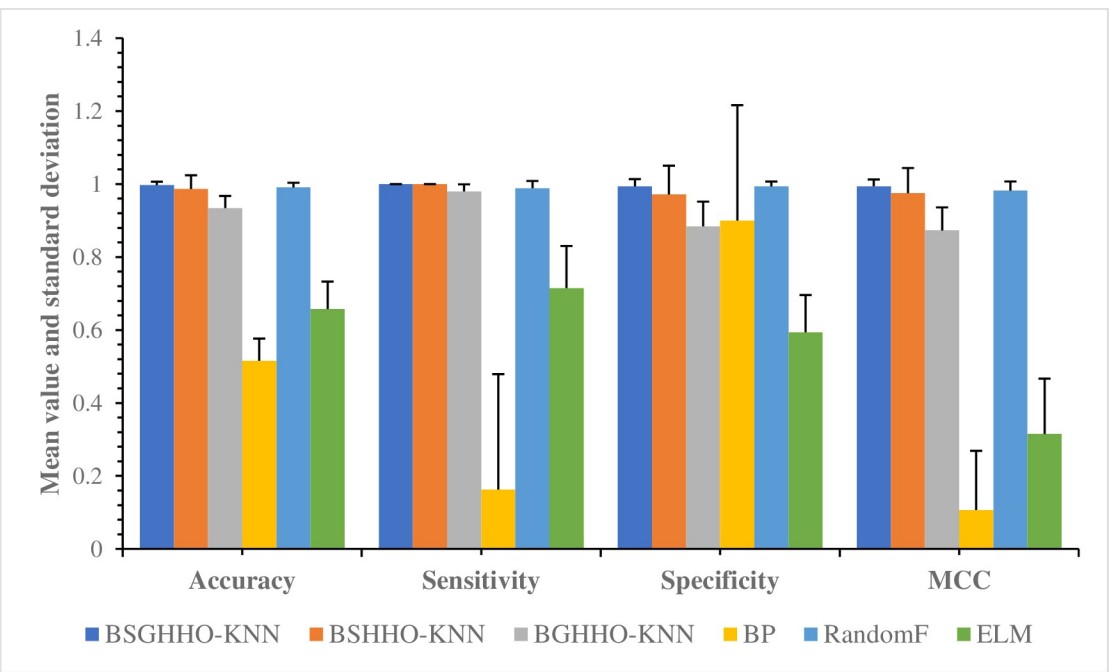

**Fig 4. Comparison of SGHHO_KNN with well-known classifiers.**

(F43) were selected most frequently. These six most common attributes appeared 10, 10, 5, 5, 5, 4, 4 and 4 times respectively. Therefore, this study concludes that these types of attributes may make a valuable contribution to predicting future trends in the sharing economy.

## 7.4. Discussion

The emergence of the sharing economy as a novel consumption model has introduced certain challenges, yet the public's expectations for its future development remain high. Through the analysis of the questionnaire experiment results, it becomes evident that numerous factors influence the future trajectory of the sharing economy. Among the 43 attributes considered,

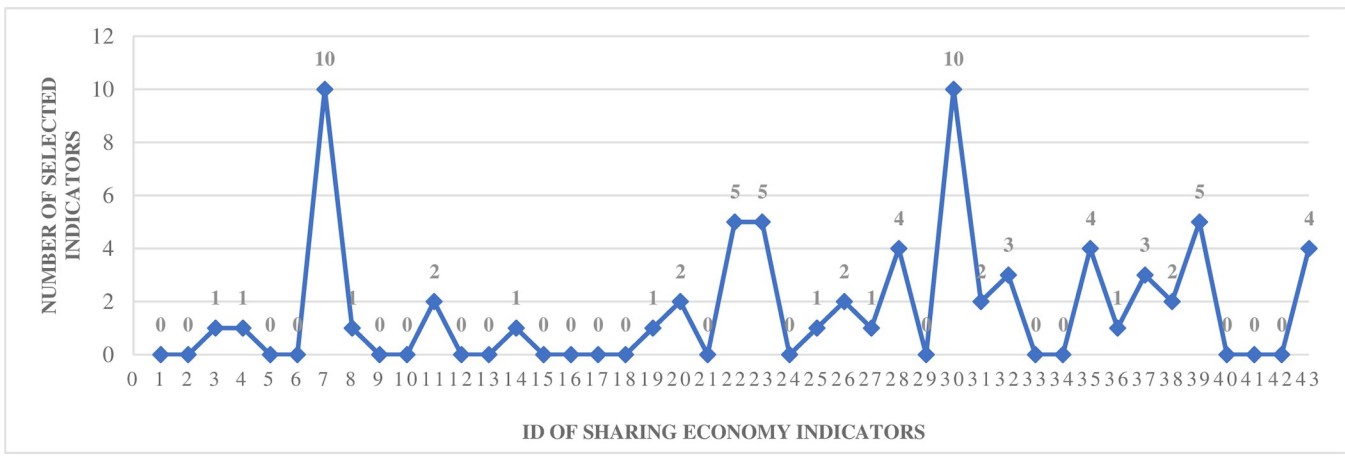

**Fig 5. The times each feature was selected by SGHHO_KNN during the 10-fold CV process.**

attributes F7, F22, F23, F28, F30, F35, F39, and F43 stand out as the most significant in shaping the sharing economy's future development trend. As a model intended to cater to the entire population, it is crucial to select the appropriate type of sharing economy that effectively serves the public, thereby playing a pivotal role in its overall advancement.

Attributes F22 and F23 pertain to consumer feedback regarding their experience with the sharing economy post-usage. In this context, the success of a particular type of sharing economy hinges on its capacity to satisfy diverse consumer needs at a reduced cost, while also garnering positive experiential feedback. This attribute plays a pivotal role in driving the development of the sharing economy. When consumers have a greater choice of products and services, providers are unable to price them effectively and consumer surplus is significantly increased. On the other hand, when the price of products and services is lower than normal, market demand increases as the price decreases, so the application of sharing economy model will help to increase market demand and thus contribute to the creation of greater value for money.

The F28, F35 and F39 belong to the troubles caused by sharing economy. Under this new model of sharing economy, the existing regulations and systems, etc. do not fit in with its development, thus problems such as taxation, labor security and information security arise one after another. Meanwhile, some of the current regulations are not yet effective in regulating the implementation of sharing economy model, and there is even the phenomenon of sharing economy model being bound by the traditional model legal system, which affects the model's value due to the lack of laws and regulations. Therefore, under the premise of clarifying the development mechanism, solving the problem of fitting the environment of sharing economy model is a key factor in developing the sharing economy.

As per the attributes F30 and F43, participants hold a crucial position in regulating the activities of the sharing economy through SGHHO. However, the current process of gathering participant feedback and opinions is inadequate, and there is a lack of focus on establishing targeted channels for collecting and incorporating their input. Add to this the fact that sharing platforms have yet to be effectively improved and perfected, leaving a lack of comprehensive, real-time regulation of participants. Therefore, encouraging public participation in the regulation of the sharing sector is a vital measure for promoting the development of sharing economy.

In summary, the sharing economy exhibits positive momentum, with a wider scope of development and an increasing variety of sharing economy products. The future growth of the sharing economy relies on the support of national policies and the continuous enhancement of the management system, which serves as a political guarantee for its stable progress. Additionally, the improvement of social infrastructure is a prerequisite for establishing and maintaining the sharing economy, while the enhancement of citizens' quality and their active and respectful involvement in the sharing economy are essential for its sustainable long-term development. Despite current challenges in the sharing economy's development, the outlook for the future remains promising.

## 8. Conclusions and future directions

In the work, we develop an effective SGHHO-KNN hybrid model to provide predictions for the future development of sharing economy. In this study, a novel variant of HHO called SGHHO is proposed. Different from other existing variants, the main contribution and innovation of the algorithm is the effective incorporation of an improved Gaussian bare bone strategy and simulated annealing mechanism. When the original HHO algorithm falls into a local optimum, the Gaussian bare-bones strategy can generate a variation factor that guides

individuals to skip from the optimal solution with a higher chance of survival. The enhanced simulated annealing mechanism enhances the exploration capability of the algorithm, enabling it to conduct more comprehensive global search operations across the entire feature space. To assess the optimization prowess of SGHHO, the IEEE CEC 2014 test suite is employed for rigorous comparative tests. The results demonstrate a substantial superiority of the SGHHO algorithm over the original HHO algorithm in 96.7% of the functions. This compelling evidence showcases the significant enhancements achieved by the SGHHO algorithm in optimizing multivariate problems, thanks to the integration of SA and GB strategies. At the same time, SGHHO significantly outperforms similar algorithms and achieves optimal solutions for 67% of the functions tested when compared to other superior algorithms. This highlights the strong competitiveness of SGHHO. Furthermore, by combining the SGHHO algorithm with KNN, a better subset of features can be obtained than previous methods. When compared to conventional machine learning approaches, the SGHHO-KNN model achieved optimal results in the evaluation metrics of ACC, sensitivity, specificity, and MCC, attaining values of 99.70%, 100.00%, 99.38%, and 99.42% respectively. The feasibility of the BSGHHO-KNN model applied to the problem of predicting future trends in the sharing economy can be fully demonstrated based on the above four evaluation indicators. In conclusion, the SGHHO-KNN approach, as proposed in this study, demonstrates its effectiveness in selecting an optimal subset of features from the available data, enabling accurate predictions of the development trend of the sharing economy.

Although the SGHHO algorithm has shown excellent performance in the above applications, there are still shortcomings and aspects that deserve further research. For instance, SGHHO does not offer a universal solution to all intricate optimization problems, and its parameters require careful consideration and analysis in a problem-specific context to achieve optimal performance. Furthermore, the SGHHO acts as a stochastic optimizer. It is randomized by nature. This means that there is still the possibility that SGHHO can fall into a local optimum in other complex applications. Therefore, the SGHHO method can also be combined with the latest optimization algorithms in future research, such as the farmland fertility algorithm, hunger games search, etc. And furthermore, it is applied in areas such as financial risk prediction and medical data diagnosis [135]. Nowadays, along with the increasing size of data in various fields, large-scale datasets also generate a large amount of redundant, useless and noisy data. This data seriously affects the performance of learning algorithms for data analysis. Thus, feature selection has an important place in this process. It is possible to drastically reduce the size of the data while maintaining the expressiveness of the information in the original feature set and thereby avoiding the combinatorial explosion problem. This is particularly true for corporate bankruptcy prediction and intelligent medical diagnosis. Therefore, the SGHHO method can be subsequently applied to the field. By combining machine learning methods to assist asset owners and medical staff in achieving intelligent decisions.

## Author Contributions

**Funding acquisition:** Hui-Ling Chen.

**Methodology:** Rongjie Li.

**Resources:** Rongjie Li, Lei Liu.

**Writing – original draft:** Qiong Wu.

**Writing – review & editing:** Xiaoxiao Tang.

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
