## [Decision Letter · Decision Letter 0]

14 Apr 2023

PONE-D-23-05468An enhanced decision-making framework for predicting future trends of sharing economyPLOS ONE

Dear Dr. Chen,

Thank you for submitting your manuscript to PLOS ONE. After careful consideration, we feel that it has merit but does not fully meet PLOS ONE’s publication criteria as it currently stands. Therefore, we invite you to submit a revised version of the manuscript that addresses the points raised during the review process.

ACADEMIC EDITOR: Major Revision

We look forward to receiving your revised manuscript.

Kind regards,

Aytaç Altan, Ph.D.

Academic Editor

PLOS ONE

Journal Requirements:

2. You indicated that ethical approval was not necessary for your study. We understand that the framework for ethical oversight requirements for studies of this type may differ depending on the setting and we would appreciate some further clarification regarding your research. Could you please provide further details on why your study is exempt from the need for approval and confirmation from your institutional review board or research ethics committee (e.g., in the form of a letter or email correspondence) that ethics review was not necessary for this study? Please include a copy of the correspondence as an "Other" file

4. Please note that PLOS ONE has specific guidelines on code sharing for submissions in which author-generated code underpins the findings in the manuscript. In these cases, all author-generated code must be made available without restrictions upon publication of the work. Please review our guidelines at https://journals.plos.org/plosone/s/materials-and-software-sharing#loc-sharing-code and ensure that your code is shared in a way that follows best practice and facilitates reproducibility and reuse.

Additional Editor Comments (if provided):

The reviewers reviewed your manuscript and expressed the opinion that some points should be handled carefully. You should revise your manuscript according to the opinions of the reviewers. In addition to the comments of the reviewers, it is useful to address the following points meticulously:

1) The readability and presentation of the study should be further improved. The paper suffers from language problems.

2) “Discussion” section should be edited in a more highlighting, argumentative way. The author should analysis the reason why the tested results is achieved. It will be helpful to the readers if some discussions about insight of the main results are added as Remarks.

3) How to set the parameters of proposed method for better performance?

4) The significance of the design carried out in this paper is not well explained relative to other important works published in this field. The authors should review, comment, and compare more works that are developed recently. The authors should clearly emphasize the contribution of the study. Please note that the up-to-date of references will contribute to the up-to-date of your manuscript. The study named " Metaheuristic optimization-based path planning and tracking of quadcopter for payload hold-release mission; Performance of metaheuristic optimization algorithms based on swarm intelligence in attitude and altitude control of unmanned aerial vehicle for path following; Artificial intelligence-based robust hybrid algorithm design and implementation for real-time detection of plant diseases in agricultural environments"- can be used to explain the object detection process in the study.

5) The main contributions of the study can be given in the last paragraph of the Introduction section.

Reviewers' comments:

Reviewer's Responses to Questions

**Comments to the Author**

1. Is the manuscript technically sound, and do the data support the conclusions?

Reviewer #1: Yes

Reviewer #2: Yes

Reviewer #3: Yes

2. Has the statistical analysis been performed appropriately and rigorously? 

Reviewer #1: Yes

Reviewer #2: Yes

Reviewer #3: Yes

3. Have the authors made all data underlying the findings in their manuscript fully available?

Reviewer #1: Yes

Reviewer #2: No

Reviewer #3: No

4. Is the manuscript presented in an intelligible fashion and written in standard English?

Reviewer #1: No

Reviewer #2: No

Reviewer #3: No

5. Review Comments to the Author

Reviewer #1: This manuscript aims to provide a reliable and intelligent prediction model for the future trend of sharing economy. It also provides a useful reference for decision-making and policy formation by relevant national authorities. Besides, the forecasting system is proposed based on an improved Harris Hawk Optimization (HHO) with a K-Nearest Neighbor (KNN) forecasting framework. The method utilizes an improved simulated annealing mechanism and a Gaussian bare bone structure to improve the original HHO, termed SGHHO. Adequate revisions to the following points should be undertaken to justify the recommendation for publication.

The abstract section is fragile. Please re-write an abstract section, explain an obtained result and contribution, improve a proposed method, etc. Please delete unnecessary information.

This paper has more than spelling and grammatical errors. Please fix all of them.

The authors should clearly state the limitations of the proposed method in other applications.

The related work section is missing; please add this section.

Please add a flowchart of the proposed method.

Please make the Introduction and related work sections more productive using the following articles. Reading and using these articles and also cited in this article: A Feature Selection Based on the Farmland Fertility Algorithm for Improved Intrusion Detection Systems, A wrapper-based feature selection for improving performance of intrusion detection systems, An improved particle swarm optimization with backtracking search optimization algorithm for solving continuous optimization problems, A multi-agent system based for solving high-dimensional optimization problems: A case study on email spam detection, Advances in Sparrow Search Algorithm: A Comprehensive Survey, An improved cuckoo search optimization algorithm with genetic algorithm for community detection in complex networks, Quantum-inspired metaheuristic algorithms: comprehensive survey and classification, An Improved Farmland Fertility Algorithm with Hyper-Heuristic Approach for Solving Travelling Salesman Problem, An Improved Harris Hawks Optimization Algorithm with Multi-strategy for Community Detection in Social Network, Slime Mould Algorithm: A Comprehensive Survey of Its Variants and Applications.

How did the authors set parameters for their proposed algorithm? Please make sensitivities of these parameters to the performance of their proposed algorithm!

All the structural problems in this study are very popular, therefore the authors are suggested to compare with results obtained from previous studies to convince the power of the proposed algorithm. Moreover, these structural problems are very easy, please try to solve other complex problems.

Please write a contribution to your paper in the Introduction section.

Please use a new comparison algorithm, such as the Farmland fertility algorithm, African Vultures Optimization Algorithm, Mountain Gazelle Optimizer, and Artificial Gorilla Troops Optimizer.

Expand the critical results in the conclusion. Focus on the main developments in the finale. Also, write the main contributions in the conclusion.

Numerical results are good enough, but more explanations are required to analyze each figure presented.

All figures have low quality, and please improve all of them.

Good luck

Reviewer #2: To provide better context on the sharing economy and its current trends, the introduction section could be expanded.

It would be beneficial to explain how the SGHHO algorithm was developed and what improvements were made compared to the original HHO algorithm.

It is better to explain is dataset public or private? if private, you need permission from data owner to be declared in paper

Author should double check similarity report of paper

While the methodology section provides a good overview of how the SGHHO-KNN model was developed, it could benefit from more detail on how the KNN classifier was used to evaluate feature subsets.

The results section clearly demonstrates how SGHHO performed in benchmark function experiments, but additional information on how these experiments were conducted would be helpful.

Although the discussion section summarizes key findings and their implications for predicting future trends in sharing economy, it could benefit from further exploration of potential limitations of this study.

It would be useful to provide practical applications of this study's findings for relevant national authorities or other stakeholders.

While the conclusion section offers a good summary of key findings and contributions, more specific recommendations for future research directions would enhance its value.

The manner of writing is typically straightforward and succinct, although certain technical jargon or ideas may necessitate additional clarification for individuals who lack familiarity with them.

Reviewer #3: The article titled “An enhanced decision-making framework for predicting future trends of sharing” has been investigated in detail. In summary, in the study, a new version of HHO is derived by integrating the Gaussian bare-bones strategy so that the algorithm does not fall into a local optimum, and the simulated annealing mechanism to improve the exploration capability to the classical HHO algorithm. With the developed algorithm, it is used to predict the future developments of the sharing economy. The topic covered in the article is remarkable and the article contains useful information. However, there are some problems that need to be addressed by the authors:

1) The introduction can be expanded to provide a clearer context for why the future of the sharing economy is be predicted using this and similar techniques

2) The authors should clearly emphasize the contribution of the study. Please note that the up-to-date of references will contribute to the up-to-date of your manuscript. The study named "Metaheuristic optimization-based path planning and tracking of quadcopter for payload hold-release mission"- can be used to explain the complexity and optimization process in the study or to indicate the contribution in the “Introduction” section.

3) Are the datasets used in the study public or private? It must be specified.

4) Does the designed optimization algorithm use a wrapper approach for the selection of indicators? It is not clearly stated in the article, I recommend that it be specified.

5) The developed algorithm is used with the KNN classifier, but there are also different algorithms used with SVM in the literature. I recommend expressing why KNN is preferred.

6) Is the algorithm developed specifically for the prediction of the sharing economy? Or can it provide high performance in different problems?

7) It would be nice if the code of the algorithm could be shared.

8) Although the Conclusion part of the study summarizes the study in general terms, this part can be expanded with some mathematical expressions in the experimental results in order to make the expression stronger.

6. PLOS authors have the option to publish the peer review history of their article (what does this mean?). If published, this will include your full peer review and any attached files.

Reviewer #1: No

Reviewer #2: No

Reviewer #3: No

---

## [Author Response · Author response to Decision Letter 0]

19 Aug 2023

Responses to all comments and revision notes

Dear Editor-In-Chief,

First, we are pleased that you offered us an invitation to revise our work. We would like to thank you for your hard work on organizing the whole reviewing process of the manuscript entitled “An enhanced decision-making framework for predicting future trends of sharing economy”

Also, we would like to express our gratitude to you, the editorial team, and the reviewers whose valuable comments on this paper have significantly improved its quality. Thanks to all anonymous reviewers for their detailed, constructive comments and suggestions, which are very helpful for us to further improve the quality of this work. We truly appreciate you assigning such qualified reviewers to our manuscript. Their efforts, visions, and insights were a tremendous help to us during this revision. 

We have addressed all the comments in the new version of the paper and will list all the changes item-by-item in response to the mentioned comments below. We have carefully answered their comments, and accordingly made revisions in the new version. 

Please note that the new modifications have been highlighted in blue in the manuscript. The details are stated as follows.

Reviewer #1: 

This manuscript aims to provide a reliable and intelligent prediction model for the future trend of sharing economy. It also provides a useful reference for decision-making and policy formation by relevant national authorities. Besides, the forecasting system is proposed based on an improved Harris Hawk Optimization (HHO) with a K-Nearest Neighbor (KNN) forecasting framework. The method utilizes an improved simulated annealing mechanism and a Gaussian bare bone structure to improve the original HHO, termed SGHHO. Adequate revisions to the following points should be undertaken to justify the recommendation for publication.

1. The abstract section is fragile. Please re-write an abstract section, explain an obtained result and contribution, improve a proposed method, etc. Please delete unnecessary information.

[Response]:

It is kind of you to give us this pertinent and amicable evaluation on this manuscript. According to your comments, we have reorganized the abstract section. The contributions and results achieved in this paper have been highlighted and unnecessary information has been removed. Thank you again for your pertinent and useful suggestions!

2. This paper has more than spelling and grammatical errors. Please fix all of them.

[Response]:

We are very sorry for the weak English writing ability of this manuscript. Following your suggestion, we have checked and corrected the entire paper. Thank you again for your comments!

3. The authors should clearly state the limitations of the proposed method in other applications.

[Response]:

It is kind of you to give us this pertinent and amicable evaluation on this manuscript. Based on your comments, we have analyzed the shortcomings of the proposed approach in the conclusion section and given areas for future development. Thank you again for your pertinent and useful suggestions!

4. The related work section is missing; please add this section.

[Response]:

Thank you for your suggestion, which is very useful for this manuscript. Following your suggestion, we have added that section to the manuscript and have studied it in two aspects: swarm intelligence algorithm and K-nearest neighbor classifier. Thank you again for your suggestion!

5. Please add a flowchart of the proposed method.

[Response]:

Thank you for your hard work on organizing the valuable suggestions on this manuscript. Based on your comments, we have not only given a flowchart of the SGHHO algorithm but also a procedure for running the SGHHO-KNN model in the manuscript. Thank you again for your valuable suggestions!

6. Please make the Introduction and related work sections more productive using the following articles. Reading and using these articles and also cited in this article: A Feature Selection Based on the Farmland Fertility Algorithm for Improved Intrusion Detection Systems, A wrapper-based feature selection for improving performance of intrusion detection systems, An improved particle swarm optimization with backtracking search optimization algorithm for solving continuous optimization problems, A multi-agent system based for solving high-dimensional optimization problems: A case study on email spam detection, Advances in Sparrow Search Algorithm: A Comprehensive Survey, An improved cuckoo search optimization algorithm with genetic algorithm for community detection in complex networks, Quantum-inspired metaheuristic algorithms: comprehensive survey and classification, An Improved Farmland Fertility Algorithm with Hyper-Heuristic Approach for Solving Travelling Salesman Problem, An Improved Harris Hawks Optimization Algorithm with Multi-strategy for Community Detection in Social Network, Slime Mould Algorithm: A Comprehensive Survey of Its Variants and Applications.

[Response]:

It is kind of you to give us this pertinent and amicable evaluation on this manuscript. Following your comments, we have read the relevant literature you provided. It has been included in the introduction to the manuscript and has further enriched the paper. Thank you again for your comments!

7. How did the authors set parameters for their proposed algorithm? Please make sensitivities of these parameters to the performance of their proposed algorithm!

[Response]:

We would like to thank you for your hard work on organizing the valuable suggestions on this manuscript and sincere apology are also expressed for you or other reviewers due to the fact that some deficiencies exist in the original manuscript objectively. The parameters involved in this study use the standard parameter settings from the original literature. And it has been shown through prior experiments that the above parameters are not sensitive to the proposed algorithm. Therefore, the experimental results are not presented in the paper. Furthermore, to maintain the integrity of the experiments. Experiments on the sensitivity of the dimensional parameters to the proposed algorithm are given in this work.

8. All the structural problems in this study are very popular, therefore the authors are suggested to compare with results obtained from previous studies to convince the power of the proposed algorithm. Moreover, these structural problems are very easy, please try to solve other complex problems.

[Response]:

It was very kind of you to give us pertinent and helpful suggestions. We searched for early published algorithms of the same type and conducted comprehensive and systematic experiments on the standard IEEE CEC 2014 Evolutionary Computation Conference testing set to validate the performance of our proposed method. Thank you again for your valuable suggestions!

9. Please write a contribution to your paper in the Introduction section.

[Response]:

It is kind of you to give us this pertinent and amicable evaluation on this manuscript. Following your comments, we have added to the introduction section. And further to highlight the contribution of this study. Thank you again for your pertinent and useful suggestions!

10. Please use a new comparison algorithm, such as the Farmland fertility algorithm, African Vultures Optimization Algorithm, Mountain Gazelle Optimizer, and Artificial Gorilla Troops Optimizer.

[Response]:

It is kind of you to give us pertinent and useful suggestions. Following your comments, we have compared the Farmland fertility algorithm with the proposed algorithm to highlight the competitive performance of the proposed algorithm. A detailed analysis of the experimental results is also given. Thank you again for your enlightening comment!

11. Expand the critical results in the conclusion. Focus on the main developments in the finale. Also, write the main contributions in the conclusion.

[Response]:

Thank you for your suggestions and it is very useful for this manuscript. Based on your comments, we have reorganized the conclusion section. Quantitative, qualitative and critical analysis has been added to this section. Moreover, the main contributions of this study are highlighted at the beginning and the ending highlights the future directions of the research. Thank you again for your comments!

12. Numerical results are good enough, but more explanations are required to analyze each figure presented.

[Response]:

Thanks for your useful suggestions. Based on your comments, we have tried to analyze each figure as detailed as possible in the manuscript, to facilitate the reader's understanding and study. Thank you again for your valuable suggestions!

13. All figures have low quality, and please improve all of them.

[Response]:

We sincerely apologize for the objective shortcomings of the original manuscript. Following your comments, we have re-uploaded all the pictures. Thank you again for your valuable suggestions!

Reviewer #2: 

1. To provide better context on the sharing economy and its current trends, the introduction section could be expanded.

[Response]:

Thank you for your suggestions and it is very useful for this manuscript. Following your comments, we have expanded the introduction by adding some background and trends related to the sharing economy. Thank you again for your suggestions!

2.It would be beneficial to explain how the SGHHO algorithm was developed and what improvements were made compared to the original HHO algorithm.

[Response]:

This issue has been dealt with in detail in our manuscript. The main contribution and innovation of the algorithm is the effective incorporation of an improved Gaussian bare bone strategy and simulated annealing mechanism. When the original HHO algorithm falls into a local optimum, the Gaussian bare-bones strategy can generate a variation factor that guides individuals to skip from the optimal solution with a higher chance of survival. The improved simulated annealing mechanism enables the algorithm to search the entire feature space more expansively by improving its exploration ability in performing global search operations.

3. It is better to explain is dataset public or private? if private, you need permission from data owner to be declared in paper.

[Response]:

The datasets analyzed during the current study available from the corresponding author on reasonable request. Thank you for your comments again!

4. Author should double check similarity report of paper

[Response]:

Thanks for your comments. Based on your comments, we have carefully compared the similarity reports of the papers. Thank you for your comments again!

5. While the methodology section provides a good overview of how the SGHHO-KNN model was developed, it could benefit from more detail on how the KNN classifier was used to evaluate feature subsets.

[Response]:

Thanks for your comments. Based on your comments, we elaborate on the operation of the KNN classifier in the theoretical section. Moreover, its basic ideas and related applications are given. Thank you for your comments again!

6. The results section clearly demonstrates how SGHHO performed in benchmark function experiments, but additional information on how these experiments were conducted would be helpful.

[Response]:

It is kind of you to give us pertinent and useful suggestions. Apologize for our shortcomings. According your suggestions, we have further described the experimental steps in this study and highlighted them in blue. Thank you again for your enlightening comment!

7. Although the discussion section summarizes key findings and their implications for predicting future trends in sharing economy, it could benefit from further exploration of potential limitations of this study.

[Response]:

Thanks for your comments and please accept my sincere apology at the same time. Based on your comments, we have analyzed the shortcomings of the proposed algorithm and the directions for future improvement in the conclusion section. Thank you for your comments again!

8. It would be useful to provide practical applications of this study's findings for relevant national authorities or other stakeholders.

[Response]:

Thanks for your comments. Based on your comments, we have presented a deeper analysis of the resulting experimental outcomes and a view on the future development of the sharing economy in the discussion section. Thank you again for your comments!

9. While the conclusion section offers a good summary of key findings and contributions, more specific recommendations for future research directions would enhance its value.

[Response]:

Thanks for your useful suggestions. Following your comments, we have given detailed guidance on the direction of the research at the end of the summary section. This includes two areas: intelligent diagnosis in medicine and bankruptcy prediction in business. Thank you again for your valuable suggestions!

10. The manner of writing is typically straightforward and succinct, although certain technical jargon or ideas may necessitate additional clarification for individuals who lack familiarity with them.

[Response]:

Thank you for your comments and please accept our sincere apologies for deficiencies. Following your comments, we have made further checks and corrections to the terminology throughout the paper. Thank you again for your comments!

Reviewer #3: 

The article titled “An enhanced decision-making framework for predicting future trends of sharing” has been investigated in detail. In summary, in the study, a new version of HHO is derived by integrating the Gaussian bare-bones strategy so that the algorithm does not fall into a local optimum, and the simulated annealing mechanism to improve the exploration capability to the classical HHO algorithm. With the developed algorithm, it is used to predict the future developments of the sharing economy. The topic covered in the article is remarkable and the article contains useful information. However, there are some problems that need to be addressed by the authors:

1. The introduction can be expanded to provide a clearer context for why the future of the sharing economy is be predicted using this and similar techniques.

[Response]:

Thank you for your suggestions and it is very useful for this manuscript. Based on your comments, we have expanded the introduction section. It also describes the importance and effectiveness of machine learning techniques to solve prediction problems. Thank you again for your comments!

2. The authors should clearly emphasize the contribution of the study. Please note that the up-to-date of references will contribute to the up-to-date of your manuscript. The study named "Metaheuristic optimization-based path planning and tracking of quadcopter for payload hold-release mission"- can be used to explain the complexity and optimization process in the study or to indicate the contribution in the “Introduction” section.

[Response]:

It is kind of you to give us pertinent and useful suggestions. Following your comments, we have highlighted the main contributions of this study in the abstract, introduction and conclusion sections respectively. In addition, relevant references have been added to enrich the article in the new manuscript. Thank you again for your comments!

3. Are the datasets used in the study public or private? It must be specified.

[Response]:

The datasets analyzed during the current study available from the corresponding author on reasonable request. Thank you for your comments again!

4. Does the designed optimization algorithm use a wrapper approach for the selection of indicators? It is not clearly stated in the article, I recommend that it be specified.

[Response]:

In this study, a wrapped feature selection method is used. Moreover, according to your comments, we have made it clear and explained in the new manuscript. Thank you again for your comments!

5. The developed algorithm is used with the KNN classifier, but there are also different algorithms used with SVM in the literature. I recommend expressing why KNN is preferred.

[Response]:

This paper combines a KNN classifier, which is a non-parametric classification technique that achieves high classification accuracy for unknown and non-normally distributed data, with many advantages such as conceptual clarity and ease of implementation. In addition, the reasons for the choice of KNN are further explained in the new manuscript. Thank you again for your comments!

6. Is the algorithm developed specifically for the prediction of the sharing economy? Or can it provide high performance in different problems?

[Response]:

In this study, the proposed algorithm demonstrated outstanding performance in predicting the future of the sharing economy. However, as a stochastic optimizer, SGHHO is inherently randomized, and there is still a possibility of falling into local optimal in other complex applications. Therefore, specific analysis is required for solving other problems. In addition, we have analyzed the deficiencies of SGHHO in the manuscript. Thank you again for your comments!

7. It would be nice if the code of the algorithm could be shared.

[Response]:

If the paper is successfully published, we will actively share the relevant code. Thank you again for your comments!

8. Although the Conclusion part of the study summarizes the study in general terms, this part can be expanded with some mathematical expressions in the experimental results in order to make the expression stronger.

[Response]:

Thank you for your comments and please accept our sincere apologies for deficiencies. Following your suggestion, in the new manuscript, we have added qualitative analysis, quantitative analysis, and critical analysis of this study to the summary section. These additions are highlighted in blue. Thank you again for your comments!

Additional Editor Comments (if provided):

1. The readability and presentation of the study should be further improved. The paper suffers from language problems.

[Response]:

We are very sorry for the weak English writing ability of this manuscript. Following your comments, we have checked and corrected the grammar of the entire paper. Thank you again for your comments!

2. “Discussion” section should be edited in a more highlighting, argumentative way. The author should analysis the reason why the tested results is achieved. It will be helpful to the readers if some discussions about insight of the main results are added as Remarks.

[Response]:

It is kind of you to give us this pertinent and amicable evaluation on this manuscript. According to your comments, we have made additions to the discussion section, particularly in-depth analysis of the key features obtained through feature selection. We also presented relevant insights on the matter. Thank you again for your pertinent and useful suggestions!

3. How to set the parameters of proposed method for better performance?

[Response]:

It is kind of you to give us this pertinent and amicable evaluation on this manuscript. The parameters involved in this study use the standard parameter settings from the original literature. And it has been shown through prior experiments that the above parameters are not sensitive to the proposed algorithm. Thank you again for your pertinent and useful suggestions!

4. The significance of the design carried out in this paper is not well explained relative to other important works published in this field. The authors should review, comment, and compare more works that are developed recently. The authors should clearly emphasize the contribution of the study. Please note that the up-to-date of references will contribute to the up-to-date of your manuscript. The study named " Metaheuristic optimization-based path planning and tracking of quadcopter for payload hold-release mission; Performance of metaheuristic optimization algorithms based on swarm intelligence in attitude and altitude control of unmanned aerial vehicle for path following; Artificial intelligence-based robust hybrid algorithm design and implementation for real-time detection of plant diseases in agricultural environments"- can be used to explain the object detection process in the study.

[Response]:

Thank you for your suggestion, which is very useful for this manuscript. Following your suggestion, we have made additions to the Introduction section. We emphasized the significance and background of this study, further highlighting its importance. Additionally, we reviewed and cited recent relevant publications to enrich the content of the paper. Thank you again for your suggestion!

5. The main contributions of the study can be given in the last paragraph of the Introduction section.

[Response]:

Thank you for your hard work on organizing the valuable suggestions on this manuscript. Following your comments, we have revised the conclusion section and highlighted the main contributions of this study. Thank you again for your valuable suggestions!

Editors: 

Dear Editor-In-Chief, 

We would like to thank you again for your hard work on organizing the whole reviewing process of the manuscript. We have tried our best to improve the manuscript and made some changes in the manuscript. These changes will not influence the content and framework of the paper. And here we did not list the changes but marked in blue in revised paper. We appreciate for Editors/Reviewers warm work earnestly and hope that the correction will meet with approval. Once again, thank you very much for your comments and suggestions!

Yours sincerely!

Corresponding author: Huiling Chen

E-mail: chenhuiling.jlu@gmail.com

---

## [Decision Letter · Decision Letter 1]

4 Sep 2023

An enhanced decision-making framework for predicting future trends of sharing economy

PONE-D-23-05468R1

Dear Dr. Chen

We’re pleased to inform you that your manuscript has been judged scientifically suitable for publication and will be formally accepted for publication once it meets all outstanding technical requirements.

Kind regards,

Salim Heddam

Academic Editor

PLOS ONE

Additional Editor Comments (optional):

Reviewer 1:The authors have completely addressed all my concerns, and I, therefore, recommend accepting this paper for publication.

Reviewer 2:The authors made the edits within the framework of the comments. Based on revision 1, the manuscript is at an acceptable level.

Reviewers' comments:

Reviewer's Responses to Questions

**Comments to the Author**

1. If the authors have adequately addressed your comments raised in a previous round of review and you feel that this manuscript is now acceptable for publication, you may indicate that here to bypass the “Comments to the Author” section, enter your conflict of interest statement in the “Confidential to Editor” section, and submit your "Accept" recommendation.

Reviewer #1: All comments have been addressed

Reviewer #3: All comments have been addressed

2. Is the manuscript technically sound, and do the data support the conclusions?

Reviewer #1: Yes

Reviewer #3: Yes

3. Has the statistical analysis been performed appropriately and rigorously? 

Reviewer #1: Yes

Reviewer #3: Yes

4. Have the authors made all data underlying the findings in their manuscript fully available?

Reviewer #1: Yes

Reviewer #3: Yes

5. Is the manuscript presented in an intelligible fashion and written in standard English?

Reviewer #1: Yes

Reviewer #3: Yes

6. Review Comments to the Author

Reviewer #1: The authors have completely addressed all my concerns, and I, therefore, recommend accepting this paper for publication.

Reviewer #3: The authors made the edits within the framework of the comments. Based on revision 1, the manuscript is at an acceptable level.

7. PLOS authors have the option to publish the peer review history of their article (what does this mean?). If published, this will include your full peer review and any attached files.

Reviewer #1: No

Reviewer #3: No

---

## [Editor Report · Acceptance letter]

27 Sep 2023

PONE-D-23-05468R1 

An enhanced decision-making framework for predicting future trends of sharing economy 

Dear Dr. Chen:

I'm pleased to inform you that your manuscript has been deemed suitable for publication in PLOS ONE. Congratulations! Your manuscript is now with our production department. 

Kind regards, 

on behalf of

Dr. Salim Heddam 

Academic Editor

PLOS ONE